# Clarification as Supervision: Reinforcement Learning for Vision-Language Interfaces

## Abstract

Recent text-only models demonstrate remarkable reasoning capabilities. Extending these to visual domains requires vision-language models to translate images into text descriptions. However, current models, trained to produce captions for human readers, often omit the precise details that reasoning systems require. This creates an interface mismatch: reasoners often fail not due to reasoning limitations but because they lack access to critical visual information. We propose Adaptive-Clarification Reinforcement Learning (AC-RL), which teaches vision models what information reasoners need through interaction. Our key insight is that clarification requests during training reveal information gaps; by penalizing success that requires clarification, we create pressure for comprehensive initial captions that enable the reasoner to solve the problem in a single pass. AC-RL improves average accuracy by 4.4 points over pretrained baselines across seven visual reasoning benchmarks, and analysis shows it would cut clarification requests by up to 39% if those were allowed. By treating clarification as a form of implicit supervision, AC-RL demonstrates that vision-language interfaces can be effectively learned through interaction alone, without requiring explicit annotations.

## 1 Introduction

Recent advances in reinforcement learning have produced text-based reasoning models with remarkable reasoning capabilities (Guo et al., 2025a; Shao et al., 2024). While these reasoning capabilities are impressive, extending them to visual domains requires careful consideration of how visual and linguistic information should interface.

Several recent works explore decoupled architectures for visual reasoning, where vision modules translate images into text descriptions that are then processed by text-only reasoners (Chen et al., 2023; Zhou et al., 2024a; Gupta & Kembhavi, 2023). This modular paradigm can offer practical advantages: it enables reuse of existing text-only reasoning models without costly multimodal retraining, allows flexible composition of specialized components, and provides interpretable interfaces between perception and reasoning. This decoupling is particularly relevant when powerful reasoners are available only through APIs that cannot be fine-tuned, or when reasoners are prohibitively expensive to fine-tune. Furthermore, many domains have specialized vision-language models (e.g., for medical imaging (Li et al., 2023), web interfaces (Lee et al., 2023), or engineering diagrams (Doris et al., 2025)) but these models often lack the broader reasoning ability of text based models. This further motivates decoupling perception from reasoning, and learning the interface between them. Systems like COLA and ViCor demonstrate this approach, using LLMs as coordinators that operate on text descriptions of visual content (Chen et al., 2024). The common thread in these approaches is that visual information flows through a linguistic bottleneck, requiring careful design of what information to communicate (Singh et al., 2024; Guo et al., 2025c).

However, this decoupling creates a critical alignment challenge: vision-language tools must learn what visual information each specific reasoner requires for successful problem-solving. Vision-language models are typically trained on diverse multimodal datasets to produce descriptions sufficient for general visual understanding and question answering. Yet differ-

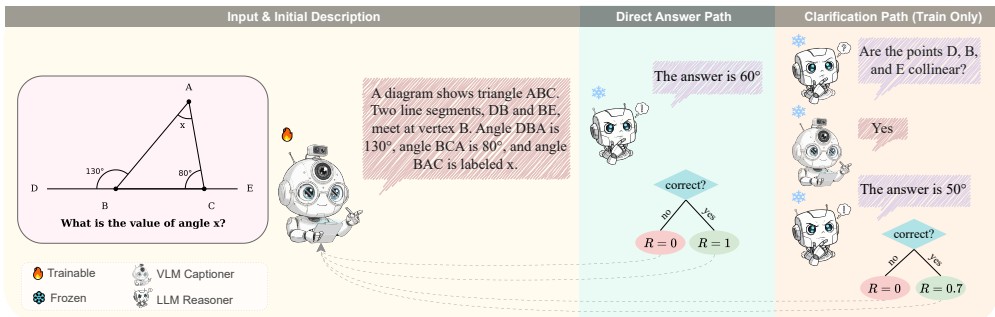

Figure 1: **Adaptive-Clarification Reinforcement Learning (AC-RL) training framework**. Given an image and a question, a trainable captioner (🔥) generates an initial description. During training, the frozen reasoner (❄) evaluates whether this description contains sufficient detail to solve the problem. If yes (Direct Answer Path), it attempts to answer directly, receiving reward $R = 1$ for correct answers or $R = 0$ for incorrect ones. If the description lacks crucial information (Clarification Path), the reasoner requests specific details, which are provided by a frozen reference captioner (❄). Correct answers after clarification receive partial reward $R = 0.7$, while incorrect answers receive $R = 0$. Gradients (dotted arrows) flow only through the initial caption generation, not through clarification responses. At inference, only the direct answer path is used: the model has learned to generate sufficiently detailed initial captions, eliminating the need for clarification.

ent reasoning models may have distinct information needs: one might excel with precise measurements, while another benefits from structural or topological descriptions. Traditional supervised approaches would require annotating "ideal captions" for each reasoner, an infeasible task given the diversity of visual reasoning problems and the implicit nature of reasoner preferences. Moreover, what constitutes an informative caption cannot be determined a priori; it emerges only through interaction with the reasoning model.

Reinforcement learning offers a natural framework for this interface learning problem, but applying it to vision-reasoner coordination is challenging. The primary difficulty lies in the sparsity of learning signals: when using binary task rewards, the vision model receives identical zero rewards whether its caption is completely inadequate or missing one crucial detail. Additionally, the large action space of language generation combined with sparse rewards leads to inefficient exploration, where most generated captions result in failure without providing informative gradients.

To overcome the limitations of this sparse reward signal, we introduce Adaptive-Clarification Reinforcement Learning (AC-RL), a framework that encourages vision–language models to produce captions aligned with the needs of a specific reasoner (Figure 1). The key idea is to make use of successful clarification during training: when the reasoner fails to solve a task directly but succeeds after a clarification exchange, this suggests that the initial caption contained partially useful information but was not complete. We assign partial credit to such clarification-based success but prefer solutions achieved on the first attempt, thereby creating optimization pressure for more informative initial captions. During training, clarification responses are generated by a frozen reference model, and gradients flow only through the initial caption. This ensures that the clarifier is not optimized and prevents the captioner from "hiding behind" a strong clarifier instead of learning to produce sufficiently informative initial captions. Over time, this leads the vision module to front-load relevant details into the initial caption, enabling single-pass inference without clarification at test time.

More specifically, AC-RL transforms sparse binary rewards into a tiered structure: full reward for direct success, partial reward (we used $\alpha = 0.7$) for success requiring clarification, and zero for failure. This densification serves dual purposes. First, it converts many zero-reward episodes into partially rewarded ones, providing a gradient signal when initial captions are nearly sufficient. Second, the penalty $(1 - \alpha)$ creates pressure to discover self-sufficient captioning strategies aligned with single-pass deployment. Through thousands of interactions, the vision model explores different description strategies, learning without ex-

plicit supervision what quantitative details, spatial relationships, or structural patterns this particular reasoner needs for solving a problem.

We evaluate AC-RL on visual reasoning benchmarks, chosen as a controlled testbed where clear ground truth enables clean measurement and strong text-only reasoners exist, isolating the vision-language interface as the primary bottleneck. Most of these benchmarks target different forms of mathematical reasoning (such as visual geometry, diagram interpretation, logic puzzles, or reasoning over plots and tables) yet some also extend beyond mathematics, including problems from chemistry, physics, and biology, e.g., MMMU (Yue et al., 2024).

The key contributions of our work are as follows:

- An exploration-based framework that enables vision-language models to discover through reinforcement learning what visual information a reasoner requires, adapting from human-caption pretraining without explicit supervision.

- A clarification-aware reward structure that uses interaction patterns as learning signals, allowing models to identify information gaps and iteratively improve their captioning strategies through trial and error.

- An empirical demonstration that our clarification-aware training scaffold effectively teaches captioners to anticipate reasoner needs, leading to improved accuracy on seven mathematical VQA benchmarks and a measurable reduction in clarification dependency at inference.

## 2 RELATED WORK

**Reinforcement learning for reasoning in language models.** Recent work shows RL can teach extended mathematical reasoning, with DeepSeek-R1 demonstrating learned policies outperform prompt-based chain-of-thought (Guo et al., 2025a). Visual extensions employ diverse strategies: training stability (Skywork-R1V2 (Wang et al., 2025b), Vision-R1 (Huang et al., 2025)), replay mechanisms (VL-Rethinker (Wang et al., 2025a), OpenVL-Thinker (Deng et al., 2025)), and cross-modal formalization (R1-OneVision (Yang et al., 2025), Mulberry (Yao et al., 2024)). These methods focus on extending reasoning chains to handle visual inputs. We take an orthogonal approach by shaping the interface between perception and reasoning modules, using clarification-aware rewards to teach captioners what information reasoners need rather than how to reason about it.

**Decoupled perception–reasoning and interface design** Decoupling visual perception from linguistic reasoning offers modularity and the ability to reuse strong text-only reasoners, but it raises an interface-alignment challenge: captions optimized for human readability may omit the quantitative and structural cues a reasoner needs (Zhou et al., 2024a; Guo et al., 2025c; Singh et al., 2024). Coordination frameworks use an LLM to route or aggregate information from one or more VLMs (e.g., COLA's coordinator that queries complementary experts) (Chen et al., 2023), or to interleave "see-think-confirm'' phases that explicitly ground and verify intermediate steps (VCTP) (Chen et al., 2024). Neuro-symbolic systems like VisProg sidestep monolithic pipelines by composing programs over off-the-shelf vision tools (Gupta & Kembhavi, 2023). Our approach adheres to the decoupled setup but replaces fixed protocols with an RL objective that learns, from interaction, which *caption features* best serve a specific reasoner.

**Learning alignment through interaction.** Several methods optimize captions specifically for reasoning rather than human readability. Most relevant to our work, RACRO directly uses binary task rewards to align a captioner to a reasoner (Gou et al., 2025), demonstrating that interface learning is possible through RL alone. However, RACRO relies solely on sparse binary rewards, which we show can be significantly improved through our clarification-aware tiered reward structure that densifies the learning signal. LAMOC and VLRM leverage language model feedback and VLM-as-reward-model, respectively (Du et al., 2023; Dzabraev et al., 2024). OmniCaptioner generates long-context descriptions that improve LLM reasoning (Lu et al., 2025), while Critic-V employs a learned VLM critic

(Zhang et al., 2025). Beyond vision, multi-agent frameworks have shown that LLMs can coordinate through language-only protocols, and that adapting inputs to a solver's biases can improve performance (Wu et al., 2024; Zhou et al., 2024b).

# 3 Methodology

## 3.1 The Vision-Reasoner Interface Problem

We consider a modular architecture where a trainable vision-language model, the captioner $\mathcal{V}_\theta$, translates images into text descriptions that enable a frozen text-only model, the reasoner $\mathcal{R}$, to solve visual reasoning tasks. Given an image $I$ and question $Q$, the system produces an answer $A$. The central challenge lies in learning what visual information the specific reasoner requires, without explicit supervision defining "ideal captions".

We specifically target scenarios where the reasoner $\mathcal{R}$ is frozen, reflecting deployment constraints where the reasoning system cannot be modified (e.g., accessible only via an API). This makes the learning problem more challenging: the captioner must unilaterally adapt to a fixed target through interaction, with no co-adaptation to make the reasoner more accommodating.

Our approach leverages successful clarification as implicit supervision. When the reasoner fails to solve a task directly but succeeds after a clarification exchange, this suggests that the initial caption was partially informative: it must have provided enough context for the reasoner to identify what might be missing, formulate a meaningful follow-up question, and ultimately solve the problem. AC-RL captures this signal through a tiered reward structure: full reward for direct success, partial reward for success after clarification, and zero otherwise. Over time, the captioner learns to anticipate the kind of information that would otherwise be requested and to include it proactively in the initial description. This leads to progressively more informative captions and enables efficient single-pass inference without clarification at test time.

## 3.2 Training and Inference Protocols

During training, we permit structured interaction between the captioner and reasoner. The captioner first generates an initial caption $c_0 \sim \pi_\theta(\cdot \mid I, Q)$ describing the visual content. The reasoner processes this caption and either produces an answer directly or requests clarification with a specific question $q_1$. When clarification is requested, a *frozen reference model* $\pi_{\text{ref}}$ provides the response $c_1 \sim \pi_{\text{ref}}(\cdot \mid I, Q, q_1)$. The reasoner then produces its final answer $A$ using all available information.

Crucially, the clarification response comes from a frozen checkpoint that receives no gradients during training. This design ensures that the captioner cannot rely on improving clarification capabilities and must instead learn to front-load relevant information into the initial caption. Details of this protocol and the complete algorithm appear in Figure 1 and Appendix A.

At inference time, the system operates in a single pass: the captioner generates one description $c_0 \sim \pi_\theta(\cdot \mid I, Q)$, and the reasoner must produce the answer based solely on this initial caption. This single-pass constraint is crucial for practical applications where multi-turn interaction would be computationally expensive or require architectural changes to existing tool-calling frameworks. By learning to front-load information during training, our approach produces captioners that work with standard single-pass inference. The reasoner processes the initial caption without needing to be modified to request clarifications.

## 3.3 Clarification-Aware Reward Design

A key contribution of AC-RL is the tiered reward structure that densifies the learning signal. In standard reinforcement learning for visual question answering, episodes receive binary rewards based solely on answer correctness. This sparse signal provides limited feedback when the captioner produces nearly sufficient but incomplete descriptions.

Our reward function addresses this sparsity by distinguishing three outcomes:

$$R(\tau) = \begin{cases} 1 & \text{if correct answer without clarification} \\ \alpha & \text{if correct answer with clarification} \\ 0 & \text{if incorrect answer} \end{cases} \quad (1)$$

where $\alpha \in (0, 1)$ and $\tau$ denotes the complete episode trajectory. We set $\alpha = 0.7$.

This structure serves dual purposes. First, it converts many zero-reward episodes into partially rewarded ones, providing gradient signal when the initial caption contains most but not all necessary information. This densification is particularly valuable early in training when captions frequently lack specific details. Second, the penalty $(1 - \alpha)$ for requiring clarification creates optimization pressure toward self-sufficient initial captions that align with single-pass deployment.

Clarification responses come from a frozen reference model rather than the training policy. This ensures gradients flow only through the initial caption, creating direct pressure to front-load information rather than rely on clarification quality. Problems beyond the reasoner's capability contribute no gradient: when all caption variants for a problem receive zero reward, all advantages are zero and the problem is effectively ignored during optimization.

The clarification mechanism thus acts as a scaffold that provides intermediate credit assignment. Episodes where the reasoner would fail with the initial caption alone but succeeds after clarification receive partial reward, signaling that the caption was nearly adequate. This graded feedback enables more sample-efficient learning compared to binary rewards that treat all failures equivalently.

### 3.4 Policy Optimization

We optimize the captioner using a KL-regularized objective that balances task performance with proximity to the pretrained initialization:

$$J(\theta) = \mathbb{E}_{(I,Q)\sim\mathcal{D}}\left[\mathbb{E}_{\tau\sim\pi_\theta}[R(\tau)]\right] - \beta \cdot D_{\mathrm{KL}}(\pi_\theta \| \pi_{\mathrm{ref}}) \quad (2)$$

where $\pi_{\mathrm{ref}}$ denotes a fixed reference policy for regularization.

We employ Beta-Normalization Policy Optimization (Xiao et al., 2025)[1], which is a variant of GRPO (Shao et al., 2024), an on-policy RL algorithm that optimize over groups of responses per prompt. Although rewards are assigned individually to each rollout $A$, the update is driven by relative performance across rollouts for the same task $(I, Q)$: only captions that perform strictly better than alternatives contribute a gradient. When all rollouts fail identically (including after clarification) no update is applied, preventing tasks unsolvable by the reasoner from penalizing otherwise potentially strong captions.

Importantly, gradients flow only through the initial caption generation $c_0$. Neither the frozen reasoner $\mathcal{R}$ nor the clarification model $\pi_{\mathrm{ref}}$ receive gradient updates, ensuring the captioner adapts unilaterally to the fixed reasoner's preferences. We prove in Appendix B that our tiered reward preserves unbiasedness of the policy gradient estimator despite post-action stochasticity from the reasoner.

## 4 Experiments

We evaluate whether Adaptive-Clarification Reinforcement Learning (AC-RL) successfully aligns vision-language models with the information needs of downstream reasoning systems. Our experimental design tests five key hypotheses: (1) AC-RL improves task performance compared to both pretrained models and standard reinforcement learning approaches (i.e., learning with the tiered rewards and clarifications is beneficial), (2) the clarification-aware training scaffold contributes meaningfully to performance gains beyond standard RL, (3) the improvements stem from learning to front-load reasoner-relevant information into initial captions, (4) AC-RL is robust to the penalty $\alpha$, and (5) generalizes to held-out reasoners.

---

[1] BNPO fits a Beta distribution to the reward distribution within each prompt group, providing more stable advantage estimation for bounded rewards compared to standard normalization. Details are provided in Appendix A

**System Architecture.** We instantiate the trainable captioning policy with InternVL3-2B or Qwen2.5-VL-3B. Their modest size enables extensive RL experimentation and ablations, and when paired with a strong reasoner, they provide reliable baseline competence across the evaluated benchmarks. Their scale also makes GRPO optimization tractable without requiring extensive computational resources. The frozen reasoning system $\mathcal{R}$ is DeepSeek-R1-Qwen-32B, a powerful text-only model trained for mathematical reasoning with particularly strong instruction following capabilities. We also evaluate the vision models as standalone systems to quantify the benefits of architectural decoupling. We chose mathematical reasoning as our evaluation domain for two reasons: (1) clear ground truth enables clean measurement of interface improvements, and (2) strong text-only reasoners exist, isolating the vision-language interface as the bottleneck rather than reasoning capability.

**Training Configurations and Method Baselines** We compare four training configurations to isolate the effects of different design choices. The **Standalone VLM** baseline has the vision-language model answer questions directly without a separate reasoner. The **Pretrained + Reasoner** configuration pairs the pretrained VLM with the frozen reasoner without fine-tuning, measuring the immediate benefit of modular architectures. **Binary-Reward RL** fine-tunes the captioner with binary task success rewards, similarly to recent work, RACRO (Gou et al., 2025). Finally, **AC-RL** employs our tiered rewards and clarification-aware training scaffold. These baselines allow us to decompose gains from architectural decoupling, reinforcement learning, and our clarification-aware training scaffold. All RL methods are trained on ViRL-39K (Wang et al., 2025a), a visual instruction dataset focused on mathematical reasoning, with evaluation performed on held-out benchmarks.

**Training Protocol.** AC-RL training uses the clarification-aware scaffold detailed in Section 3. During training, the captioner generates $c_0 \sim \pi_\theta$, and if the reasoner requests clarification, a frozen reference policy provides the response. The tiered rewards ($R = 1$ for direct success, $R = 0.7$ with clarification, $R = 0$ for failure) create gradients only through the initial caption. We optimize using BNPO with KL regularization. Notably, AC-RL maintains greater generation diversity than standard RL throughout training (Appendix D).

**Evaluation Protocol.** All models are evaluated using **single-pass evaluation**: the captioner produces a description that the reasoner uses to generate a final answer, with no clarification permitted. This protocol ensures that performance gains reflect improved caption quality rather than multi-turn interaction benefits. For behavioral analyses in Section 4.3, we additionally conduct instrumented runs with **clarification-enabled evaluation** where clarification is allowed, to measure clarification patterns. We evaluate on seven benchmarks spanning diverse visual math reasoning challenges and compare against leading proprietary and open-weights models (details in Appendix C). We report exact-match accuracy using EvalScope (Team, 2024a) and VLMEvalKit (Duan et al., 2024).

## 4.1 Overall Performance

Table 1 presents our results in the context of leading proprietary and open-weights models. We first note that small vision-language models achieve limited performance when solving problems directly: InternVL-2B and Qwen2.5-VL-3B reach only 32.4% and 34.6% average accuracy respectively as standalone systems. Simply pairing these models with a strong reasoner (Pretrained + Reasoner) improves performance to 39.3% and 39.0%, demonstrating the value of modular architectures. However, applying AC-RL yields the most gains.

With a Qwen2.5-VL-3B captioner, AC-RL improves the average accuracy from 39.0 to 43.4 (+4.4 points), with substantial gains on robustness and vision-centric benchmarks like *Dyna-Math* (+10.6) and *MathVerse* (+5.2). The InternVL-2B captioner sees a similar +3.3 average point increase. These results, obtained under an identical single-pass protocol, demonstrate that AC-RL effectively aligns the captioning policy with the downstream reasoner's needs.

While we observe minor regressions on WeMath ($-0.7$ to $-1.5$ points), this benchmark explicitly targets preexisting knowledge deficits rather than visual extraction. AC-RL's optimization pressure prioritizes precise visual details, yielding substantial gains on extraction-

heavy tasks like DynaMath and MathVerse. AC-RL excels at enabling reasoners to see more, but cannot address the reasoner's internal knowledge gaps such as missing formulas, which limit WeMath performance.

Table 1: Main results on multi-modal reasoning benchmarks: MathVista (MVista), Math-Vision (MVision), MathVerse (MVerse), MMMU, WeMath (WeM), DynaMath (DynaM), and LogicVista (LVista). Our AC-RL method, evaluated in the final blocks for each model size, significantly enhances the performance of small vision models.

| Model | MVista | MVision | MVerse | MMMU | WeM | DynaM | LVista | AVG |
|---|---|---|---|---|---|---|---|---|
| **Proprietary Models** | | | | | | | | |
| GPT-4o-20241120 | 60.0 | 31.2 | 40.6 | 70.7 | 45.8 | 34.5 | 52.8 | 47.9 |
| Gemini-2.0-Flash | 70.4 | 43.6 | 47.7 | 72.6 | 47.4 | 42.1 | 52.3 | 53.7 |
| Claude-3.7-Sonnet | 66.8 | 41.9 | 46.7 | 75.0 | 49.3 | 39.7 | 58.2 | 53.9 |
| o1 | 73.9 | 42.2 | — | **78.2** | — | — | — | — |
| Gemini 2.5 Pro | **80.9** | **69.1** | **76.9** | 74.7 | **78.0** | **56.3** | **73.8** | **72.8** |
| Seed1.5-VL (Thinking) | 79.5$^\dagger$ | 68.7 | — | 77.9 | 77.5 | — | — | 75.9$^*$ |
| **Open-Weights Models** | | | | | | | | |
| InternVL3-2B-MPO | 57.0 | 21.7 | 25.3 | 48.6 | 22.4 | 14.6 | 36.9 | 32.4 |
| InternVL3-8B-MPO | 71.6 | 29.3 | 39.8 | 62.7 | 37.1 | 25.5 | 44.1 | 44.3 |
| Ovis2-8B | 71.8$^\dagger$ | 25.9 | 42.3 | 59.0 | — | — | 39.4 | 47.7 |
| InternVL3-14B-MPO | 75.1 | 37.2 | 44.4 | 67.1 | 43.0 | 31.3 | 51.2 | 49.9 |
| QVQ-72B-Preview | 70.3 | 34.9 | 48.2 | 70.3 | 39.0 | 30.7 | 58.2 | 50.2 |
| MMR1-Math-v0-7B | 71.0$^\dagger$ | 30.2 | 49.2 | — | — | — | 50.8 | 50.3 |
| InternVL3-38B-MPO | 75.1 | 34.2 | 48.2 | 70.1 | **48.6** | **35.3** | **58.4** | 52.8 |
| VL-Rethinker-72B | **80.3** | **43.9** | — | 68.8 | — | — | — | — |
| InternVL3-78B-MPO | 79.0 | 43.1 | **51.0** | **72.2** | 46.0 | 35.1 | 55.9 | **54.6** |
| **InternVL-2B** | | | | | | | | |
| Standalone VLM | 57.0 | 21.9 | 25.3 | 48.6 | 22.4 | 14.6 | 36.9 | 32.4 |
| Pretrained + Reasoner | 61.0 | 34.7 | 28.9 | 57.4 | **32.8** | 12.0 | 48.3 | 39.3 |
| **AC-RL (ours)** | **65.3** | **36.7** | **36.8** | **58.4** | 32.1 | **20.0** | **49.0** | **42.6** |
| | (+4.3) | (+2.0) | (+7.9) | (+1.0) | (-0.7) | (+8.0) | (+0.7) | (+3.3) |
| **Qwen2.5-VL-3B** | | | | | | | | |
| Standalone VLM | **64.5** | 21.9 | 28.8 | 50.1 | 24.2 | 13.4 | 39.6 | 34.6 |
| Pretrained + Reasoner | 59.7 | 32.8 | 29.2 | 55.2 | **34.7** | 14.2 | 47.2 | 39.0 |
| **AC-RL (ours)** | 63.8 | **36.8** | **34.4** | **57.7** | 33.2 | **24.8** | **53.0** | **43.4** |
| | (+4.1) | (+4.0) | (+5.2) | (+2.5) | (-1.5) | (+10.6) | (+5.8) | (+4.4) |

$\dagger$ Result on *testmini/mini* subset.

## 4.2 Ablations

To better understand the source of these improvements, we analyze the incremental value of each component in our approach using the Qwen2.5-VL-3B model (Table 2). All configurations are evaluated using direct inference (no clarification allowed).

### 4.2.1 Decomposition of Performance Gains

Table 2: **Decomposition of performance gains on Qwen2.5-VL-3B** across multimodal reasoning benchmarks: MathVista (MVista), MathVision (MVision), MathVerse (MVerse), MMMU, WeMath (WeM), DynaMath (DynaM), and LogicVista (LVista). All models are evaluated in a single-pass setting. The results show that AC-RL provides a significant performance boost beyond both architectural decoupling and Binary Rewards.

| Training Method | MVista | MVision | MVerse | MMMU | WeM | DynaM | LVista | AVG |
|---|---|---|---|---|---|---|---|---|
| VLM-only (No Reasoner) | **64.50** | 21.90 | 28.80 | 50.10 | 24.20 | 13.40 | 39.60 | 34.64 |
| Decoupled (No RL) | 59.69 | 32.80 | 29.18 | 55.22 | **34.71** | 14.17 | 47.20 | 39.00 |
| Binary-Reward RL | 62.60 | 34.30 | 31.09 | 55.44 | 33.45 | 17.56 | 47.42 | 40.27 |
| **AC-RL (Ours)** | 63.80 | **36.84** | **34.39** | **57.70** | 33.22 | **24.75** | **53.02** | **43.39** |

The results illustrate a clear progression. First, decoupling the captioner from the reasoner (Pretrained + Reasoner) yields substantial gains, particularly on structurally complex tasks like *MathVision* ($21.9 \rightarrow 32.8$). Second, applying binary-reward RL provides further improvements across most benchmarks. However, our clarification-aware AC-RL delivers the most substantial gains over Binary-Reward RL: *DynaMath* improves by +7.2 points ($17.56 \rightarrow 24.75$), *LogicVista* by +5.6 points, and *MathVerse* by +3.3 points.

The improvements on DynaMath are especially noteworthy. While Binary-Reward RL achieves modest gains over the pretrained baseline (+3.4 points), AC-RL delivers an additional +7.2 points, reaching 24.75% accuracy on the most challenging problem variants.

### 4.2.2 SENSITIVITY TO CLARIFICATION PENALTY

The clarification penalty $(1 - \alpha)$ controls the reward reduction for success requiring clarification. Table 3 shows results across penalty values $\{0.1, 0.3, 0.5, 0.7\}$. Intermediate penalties (0.3, 0.5) consistently outperform extremes, with near-identical MathVision performance (40.2 vs 40.1) demonstrating robustness. Penalties that are too weak (0.1) provide insufficient optimization pressure, while too strong (0.7) approaches binary rewards, losing partial credit benefits.

Table 3: Sensitivity to the clarification penalty $(1 - \alpha)$.

| Penalty | LVista | MVision |
|---------|--------|---------|
| 0.1     | 53.2   | 35.8    |
| **0.3** | **55.7** | **40.2** |
| 0.5     | 52.4   | 40.1    |
| 0.7     | 50.1   | 35.5    |

### 4.2.3 GENERALIZATION ACROSS REASONERS

A natural question is whether learned strategies generalize or overfit to the training reasoner. Table 4 compares captioners paired with DeepSeek-R1-0528-Qwen3-8B, differing in size and checkpoint from our training reasoner (DeepSeek-R1-Qwen-32B). With the training reasoner, AC-RL improves by +8.5 and +5.5 points on Log-

Table 4: Cross-reasoner generalization.

| Captioner  | Reasoner    | LVista | MVision |
|------------|-------------|--------|---------|
| Pretrained | Train (32B) | 47.2   | 34.7    |
| AC-RL      | Train (32B) | **55.7** | **40.2** |
| Pretrained | Unseen (8B) | 44.3   | 30.0    |
| AC-RL      | Unseen (8B) | 53.0   | 36.5    |

icVista and MathVision. With the unseen reasoner, AC-RL achieves comparable gains (+8.7, +6.5), suggesting it learns generally useful captioning strategies rather than exploiting idiosyncrasies of the training reasoner.

### 4.3 ANALYSIS OF INTERFACE BEHAVIOR

To understand the mechanism underlying AC-RL's performance gains, we analyze how the training procedure modifies the captioner's behavior. Our hypothesis is that the clarification-aware reward teaches the model to *front-load* reasoner-salient information into the initial caption, thereby reducing the need for clarification.

We first measure the **clarification attempt rate** using clarification-enabled evaluation (for measurement purposes only) and counting how frequently it requests clarification when processing captions from AC-RL-trained versus baseline models. Table 5 shows that AC-RL dramatically reduces the

Table 5: Clarification attempt rate (%).

|               | MathVision | MathVerse |
|---------------|------------|-----------|
| Binary Reward | 40.7       | 49.6      |
| AC-RL         | 29.0       | 30.3      |
| **Reduction** | **29%**    | **39%**   |

frequency of clarification requests. On MathVision, the clarification rate drops from 40.69% (Binary-Reward RL baseline) to 28.95% (AC-RL). On MathVerse, the reduction is even more pronounced at 39%. This confirms that AC-RL-trained captioners learn to preemptively include information that would otherwise trigger follow-up questions.

Building on this evidence, we compute the **clarification gap**: the difference in accuracy between clarification-enabled evaluation and single-pass evaluation. A smaller gap indicates that the initial caption is more informationally self-sufficient. Table 6 presents these results. For the baseline model, allowing clarification provides substantial accuracy gains:

+2.89 points on MathVision and +4.06 points on MathVerse. In contrast, the AC-RL model shows minimal benefit from clarification and even a negative gap on MathVerse (-2.54), suggesting that its initial captions are so well-aligned that additional clarification can sometimes introduce noise. The relative improvement metric (fraction of previously incorrect answers that become correct with clarification) further confirms this pattern: AC-RL achieves 1.5% relative improvement on MathVision versus 4.4% for the baseline.

Table 6: Performance gap between clarification-enabled and single-pass evaluation. "Rel" denotes the fraction of previously incorrect answers that become correct when clarification is allowed. Smaller gaps indicate greater self-sufficiency of initial captions.

| Dataset | Model | Clarification-Enabled | Single-Pass | Gap (Abs / Rel) |
|---|---|---|---|---|
| MathVision | AC-RL | 37.66% | 36.71% | **+0.95 / 0.015** |
| | Binary-Reward | 37.20% | 34.31% | +2.89 / 0.044 |
| MathVerse_MINI | AC-RL | 34.26% | 36.80% | **-2.54 / -0.040** |
| | Binary-Reward | 35.15% | 31.09% | +4.06 / 0.059 |

Finally, to assess whether clarification requests are genuinely necessary, we measure accuracy under **denied clarification**: we identify instances where the model requested clarification during clarification-enabled evaluation, then examine the single-pass accuracy on this same subset (equivalent to denying the clarification request). The drop $\Delta_{\text{deny}} = \text{Acc}_{\text{clarification-enabled}} - \text{Acc}_{\text{denied}}$ quantifies how much the model relies on clarification when it requests it. Table 7 shows that while AC-RL reduces overall clarification frequency, its remaining requests are more selective. The AC-RL model exhibits a larger performance drop when clarification is denied ($\Delta_{\text{deny}} = 14.21$ on MathVision versus 11.43 for the baseline), despite making fewer requests overall (880 versus 1,237). This suggests that AC-RL learns to distinguish between recoverable and irrecoverable information gaps: it produces self-sufficient captions when possible, but when it does request clarification, these requests target instances where critical visual details cannot be inferred from context alone.

Table 7: Accuracy impact of denying clarification on instances where it was requested. $\text{Acc}_{\text{deny}}$ is computed on the subset of problems where the reasoner requested clarification during clarification-enabled evaluation; for these specific problems, we measure accuracy using only the initial caption (i.e., denying the clarification request).

| Dataset | Model | # Requests | $\text{Acc}_{\text{single-pass}}$ | $\text{Acc}_{\text{deny}}$ / $\Delta_{\text{deny}}$ |
|---|---|---|---|---|
| MathVision | AC-RL | 880 | 36.71% | 22.50% / **14.21** |
| | Binary-Reward | 1237 | 34.31% | 22.88% / 11.43 |
| MathVerse_MINI | AC-RL | 276 | 36.80% | 33.33% / **3.47** |
| | Binary-Reward | 496 | 31.09% | 28.43% / 2.66 |

## 4.4 SUBJECT-LEVEL PERFORMANCE ANALYSIS

To better understand the nature of these improvements, we conduct a fine-grained analysis of performance across different mathematical subjects and difficulty levels. This reveals whether the model is generically improving or learning to prioritize specific types of information relevant to the reasoner. We decompose the MathVision and DynaMath benchmarks by subject area and compute per-subject accuracy for both the AC-RL model and the pretrained baseline (Qwen2.5-VL-3B + Reasoner configuration). Additionally, we analyze DynaMath average performance stratified by education level (elementary, high school, undergraduate) to assess whether AC-RL's benefits vary with problem complexity.

Figure 2 visualizes the per-subject performance comparison. The analysis reveals that AC-RL's gains are concentrated in subjects that depend heavily on precise quantitative and structural information. On MathVision, we observe the largest improvements in metric geometry for angles (+10.4 points), transformation geometry (+8.3 points), and algebra (+7.5 points). DynaMath shows even more pronounced gains in solid geometry (+18.7 points), algebra (+15.1 points), and puzzle tests (+13.5 points). These subjects arguably

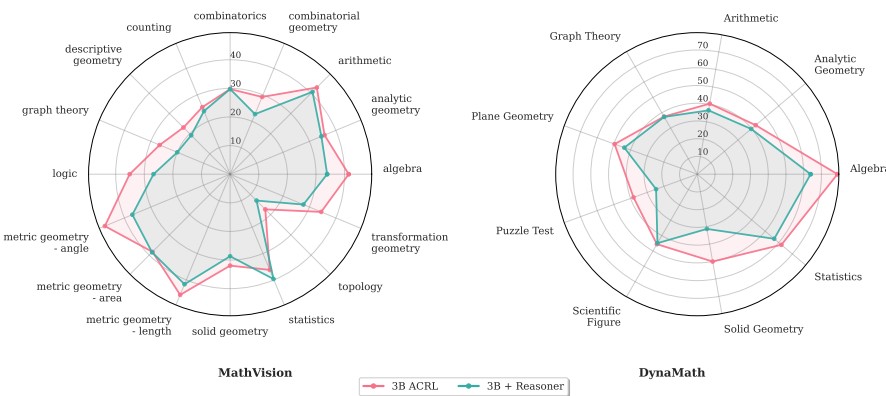

Figure 2: Subject-level performance comparing AC-RL to the pretrained baseline using Qwen2.5-VL-3B + Reasoner. Left: MathVision subjects. Right: DynaMath categories. AC-RL shows targeted improvements in quantitatively-intensive.

require extracting specific numerical values, spatial relationships, or structural patterns from images. In contrast, performance differences are minimal in subjects that rely more on general visual understanding or pattern recognition.

Figure 3 shows that AC-RL maintains consistent improvements across all difficulty levels on DynaMath. The absolute gains remain relatively stable at 6.5, 6.3, and 4.3 points for elementary, high school, and undergraduate levels, respectively. While both models show expected degradation as problem complexity increases, AC-RL preserves its advantage by learning to extract critical visual details needed at each level. The smaller gain at the undergraduate level may reflect inherent limits in what visual information alone can contribute to abstract problems.

This non-uniform improvement pattern indicates that AC-RL learns to extract and prioritize the specific types of information most valuable to the downstream reasoner. The selective nature of these improvements suggests that the clarification-aware training identifies and addresses systematic information gaps in the original captioning policy, with the model discovering domain-specific extraction strategies through interaction rather than explicit supervision.

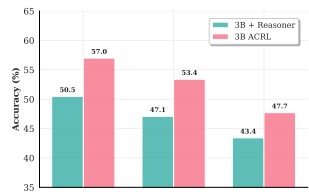

Figure 3: DynaMath average accuracy across education levels. AC-RL consistently outperforms the baseline regardless of problem difficulty.

## 5 CONCLUSION

We presented Adaptive-Clarification Reinforcement Learning (AC-RL), a framework that learns vision-reasoner interfaces through interaction rather than supervision. By using clarification requests as implicit feedback and tiered rewards, AC-RL enables captioners to discover what information their paired reasoner requires without explicit annotation. Our experiments demonstrate consistent improvements across seven mathematical reasoning benchmarks, with particularly strong gains on quantitatively-intensive domains.

The success of AC-RL suggests that interface alignment between AI modules can be learned through reinforcement learning without requiring explicit caption annotations. We demonstrated this on mathematical reasoning as a controlled testbed where clear ground truth enables clean measurement. The core mechanism of penalizing clarification-dependent success to encourage information front-loading could naturally extend to other settings where specialized VLMs interface with frozen reasoners, such as medical imaging (Li et al., 2023) or engineering diagrams (Doris et al., 2025); we leave empirical validation of such extensions for future work. Other promising directions include bidirectional adaptation where both modules co-evolve, using clarification content (not just occurrence) as richer supervision, and multi-turn clarification with decaying rewards for iterative refinement.

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

# A  AC-RL Algorithm

We provide a formal specification of the Adaptive-Clarification Reinforcement Learning (AC-RL) algorithm. The algorithm operates in an episodic setting where each episode consists of a visual reasoning problem $(I, Q)$ sampled from the dataset $\mathcal{D}$.

## A.1  Formal Problem Setup

Let $\mathcal{M} = (\mathcal{S}, \mathcal{A}, P, R, \gamma)$ denote the Markov Decision Process where:

- $\mathcal{S} = \mathcal{I} \times \mathcal{Q} \times \mathcal{H}$ is the state space,
- $\mathcal{A} = \mathcal{C}$ is the action space (caption generation),
- $P : \mathcal{S} \times \mathcal{A} \to \Delta(\mathcal{S})$ is the transition kernel,
- $R : \mathcal{T} \to [0, 1]$ is the reward function defined on trajectories,
- $\gamma = 1$ (undiscounted episodic setting).

A single episode proceeds as follows. At $t = 0$, the vision policy emits the initial caption $c_0 \sim \pi_\theta(\cdot \mid s_0)$ with $s_0 = (I, Q, \emptyset)$. The reasoner stochastically decides whether to request clarification and, if so, which question to ask; we denote this by a $\theta$-independent kernel $q_1 \sim p(\cdot \mid s_0, c_0)$. When $q_1 \neq \emptyset$, the clarification caption is produced by a *frozen* checkpoint $\pi_{\text{ref}}$:

$$c_1 \sim \pi_{\text{ref}}(\cdot \mid s_1), \qquad s_1 = (I, Q, \{c_0, q_1\}),$$

and the reasoner produces a final answer according to a $\theta$-independent kernel $A \sim p(\cdot \mid Q, c_0, (q_1, c_1))$. When $q_1 = \emptyset$, the reasoner answers from $(Q, c_0)$ directly. The next state appends the sampled variables to the dialogue history. Thus $P$ composes the reasoner's stochastic behavior and the frozen clarification-caption policy $\pi_{\text{ref}}$; conditioned on the agent's action $c_0$, these post-action mechanisms are $\theta$-*independent* by construction. The episode terminates after the answer $A$ is produced, and the reward is assigned as in the main text.

**Clarification captioning is frozen.** In all experiments, the clarification caption $c_1$ is generated by a *frozen* checkpoint $\pi_{\text{ref}}$ (typically the reference policy). Its distribution does not change during training. Consequently, no gradients flow through $\pi_{\text{ref}}$ or through the reasoner $\mathcal{R}$; only the log-probabilities of the initial caption tokens $c_0$ contribute to the policy update.

## A.2  Advantage Computation

We use Beta-Normalization Policy Optimization (BNPO) (Xiao et al., 2025) for advantage estimation. BNPO addresses a limitation of standard GRPO: while GRPO uses fixed normalization, BNPO adaptively normalizes rewards using a Beta distribution whose parameters evolve with the policy. This provides lower-variance gradient estimates and more stable training.

For a group of rewards $\{R^{(i)}\}_{i=1}^M$ from a single prompt, BNPO fits Beta distribution parameters $(\alpha_\beta, \beta_\beta)$ via method-of-moments from the group statistics, then computes advantages as:

$$A_{\text{BNPO}}^{(i)} = \frac{R^{(i)} - \mu_\beta}{\sigma_\beta + \epsilon}, \quad \text{where } \mu_\beta = \frac{\alpha_\beta}{\alpha_\beta + \beta_\beta} \tag{3}$$

Although BNPO was designed for binary rewards, we found it effective with our ternary reward structure, moderately outperforming standard GRPO in our experiments.

## A.3  Policy Update

The policy is updated using the clipped surrogate objective with a fixed KL reference:

$$\mathcal{L}_{\text{clip}}(\theta) = -\mathbb{E}_{(s_t, a_t)} \left[ \min \left( r_t(\theta) A_t, \text{clip}(r_t(\theta), 1 - \epsilon, 1 + \epsilon) A_t \right) \right] + \beta \, D_{KL}(\pi_\theta \| \pi_{\text{ref-KL}}), \tag{4}$$

where $r_t(\theta) = \pi_\theta(a_t \mid s_t)/\pi_{\theta_{\text{old}}}(a_t \mid s_t)$, and $A_t$ is the advantage computed via BNPO (Xiao et al., 2025). The gradient is computed solely on the initial captioning segments $c_0$; the clarification responses $c_1$ are emitted by the frozen $\pi_{\text{ref}}$ and are thus $\theta$-independent.

## A.4 Training Algorithm

---

**Algorithm 1** Adaptive-Clarification Reinforcement Learning (AC-RL)

---

**Require:** Dataset $\mathcal{D}$, vision model $\mathcal{V}_\theta$, reasoner $\mathcal{R}$, penalty $\alpha$, group size $M$, gradient steps $K$

1: Initialize policy $\pi_\theta \leftarrow \mathcal{V}_\theta$ with parameters $\theta$
2: Initialize *frozen* clarification captioner $\pi_{\text{ref}}$ (checkpoint used only for $c_1$)
3: Initialize fixed KL reference $\pi_{\text{ref}} \leftarrow \pi_\theta$
4: **for** iteration $t = 1$ to $T$ **do**
5:    Sample batch $\mathcal{B} = \{(I_j, Q_j)\}_{j=1}^B \sim \mathcal{D}$
6:    **for** each $(I_j, Q_j) \in \mathcal{B}$ **do**
7:       **for** $i = 1$ to $M$ **do**
8:          Generate initial caption: $c_0^{(i,j)} \sim \pi_\theta(\cdot \mid I_j, Q_j)$
9:          Sample reasoner's clarification decision: $q_1^{(i,j)} \sim \mathcal{R}_{\text{clarify}}(\cdot \mid Q_j, c_0^{(i,j)})$     (no gradients)
10:         **if** $q_1^{(i,j)} \neq \emptyset$ **then**
11:            Generate clarification caption from *frozen* checkpoint: $c_1^{(i,j)} \sim \pi_{\text{ref}}(\cdot \mid I_j, Q_j, (c_0^{(i,j)}, q_1^{(i,j)}))$     (no gradients)
12:            Get answer: $A^{(i,j)} \sim \mathcal{R}(\cdot \mid Q_j, c_0^{(i,j)}, (q_1^{(i,j)}, c_1^{(i,j)}))$     (no gradients)
13:            Set clarification flag: $C^{(i,j)} = 1$
14:         **else**
15:            Get answer: $A^{(i,j)} \sim \mathcal{R}(\cdot \mid Q_j, c_0^{(i,j)})$     (no gradients)
16:            Set clarification flag: $C^{(i,j)} = 0$
17:         **end if**
18:         Compute reward: $R^{(i,j)} = \begin{cases} 1.0 & \text{if } \text{correct}(A^{(i,j)}) \wedge C^{(i,j)} = 0 \\ \alpha & \text{if } \text{correct}(A^{(i,j)}) \wedge C^{(i,j)} = 1 \\ 0 & \text{otherwise} \end{cases}$
19:       **end for**
20:       Fit Beta parameters $(\alpha_\beta^{(j)}, \beta_\beta^{(j)})$ to $\{R^{(i,j)}\}_{i=1}^M$
21:       Compute BNPO advantages $\{A_{\text{BNPO}}^{(i,j)}\}_{i=1}^M$
22:    **end for**
23:    **for** $k = 1$ to $K$ **do**
24:       Update policy with clipping and KL penalty: $\theta \leftarrow \theta - \eta \nabla_\theta \mathcal{L}_{\text{clip}}(\theta; \pi_{\text{ref-KL}})$
25:    **end for**
26: **end for**
27: **return** Trained policy $\pi_\theta$

---

## B  Unbiasedness of the Three-Tier Reward

In this section, we provide a formal proof that our three-tier reward structure maintains the unbiasedness property of the REINFORCE policy gradient estimator, even when the reasoner exhibits stochasticity.

**Theorem 1** (Unbiasedness of the Three-Tier Reward with Stochastic Reasoner). *Let $\xi \sim p(\cdot \mid \tau)$ denote all post-action randomness after the policy chooses its actions (e.g., the reasoner's sampling noise and, when clarification is used, the frozen clarification-caption sampling). Define the extended trajectory $\tilde{\tau} = (\tau, \xi)$ with joint density:*

$$p_\theta(\tilde{\tau}) = p_\theta(\tau) \cdot p(\xi \mid \tau) \tag{5}$$

*where $p(\xi \mid \tau)$ is independent of $\theta$.*

*Let the tiered reward function be defined as:*

$$R_{tier}(\tilde{\tau}) = \begin{cases} 1 & \text{if } correct(A(\tilde{\tau})) \wedge C(\tilde{\tau}) = 0 \\ \alpha & \text{if } correct(A(\tilde{\tau})) \wedge C(\tilde{\tau}) > 0 \\ 0 & \text{otherwise} \end{cases} \tag{6}$$

*where $\alpha \in (0,1)$, and define the training objective:*

$$J(\theta) = \mathbb{E}_{\tilde{\tau} \sim p_\theta}[R_{tier}(\tilde{\tau})]. \tag{7}$$

*For any baseline $b_t(s_t)$ that does not depend on the action $a_t$, the REINFORCE estimator:*

$$\hat{g}(\tilde{\tau}) = \sum_{t=0}^{T-1} \nabla_\theta \log \pi_\theta(a_t \mid s_t) \cdot (R_{tier}(\tilde{\tau}) - b_t(s_t)) \tag{8}$$

*satisfies $\mathbb{E}_{\tilde{\tau} \sim p_\theta}[\hat{g}(\tilde{\tau})] = \nabla_\theta J(\theta)$, i.e., the policy gradient remains unbiased despite the $0/\alpha/1$ reward shaping and post-action stochasticity.*

*Proof.* **Step 1: Setup.** The extended trajectory $\tilde{\tau} = (\tau, \xi)$ includes both the policy-generated trajectory $\tau = (s_0, a_0, s_1, a_1, ..., s_T)$ and the post-action randomness $\xi$. The joint probability decomposes as:

$$p_\theta(\tilde{\tau}) = p(s_0) \prod_{t=0}^{T-1} \pi_\theta(a_t \mid s_t) \, P(s_{t+1} \mid s_t, a_t) \cdot p(\xi \mid \tau), \tag{9}$$

where $p(s_0)$ is the initial state distribution, $P$ is the environment transition kernel, and $p(\xi \mid \tau)$ is the distribution over post-action randomness given the trajectory; by assumption, $p(\xi \mid \tau)$ is $\theta$-independent.

**Step 2: Policy Gradient Theorem.** For $R_{\text{tier}}(\tilde{\tau})$,

$$\nabla_\theta J(\theta) = \nabla_\theta \int p_\theta(\tau) \, p(\xi \mid \tau) \, R_{\text{tier}}(\tilde{\tau}) \, d\xi \, d\tau \tag{10}$$

$$= \int p_\theta(\tau) \, p(\xi \mid \tau) \, R_{\text{tier}}(\tilde{\tau}) \, \nabla_\theta \log p_\theta(\tau) \, d\xi \, d\tau \tag{11}$$

$$= \int p_\theta(\tilde{\tau}) \, R_{\text{tier}}(\tilde{\tau}) \, \nabla_\theta \log p_\theta(\tau) \, d\tilde{\tau}, \tag{12}$$

using that $\nabla_\theta \log p_\theta(\tilde{\tau}) = \nabla_\theta \log p_\theta(\tau) + \nabla_\theta \log p(\xi \mid \tau)$ and $\nabla_\theta \log p(\xi \mid \tau) = 0$ by $\theta$-independence. Since $p(s_0)$ and $P$ are $\theta$-independent,

$$\nabla_\theta \log p_\theta(\tau) = \sum_{t=0}^{T-1} \nabla_\theta \log \pi_\theta(a_t \mid s_t), \tag{13}$$

hence

$$\nabla_\theta J(\theta) = \mathbb{E}_{\tilde{\tau} \sim p_\theta} \left[ \sum_{t=0}^{T-1} \nabla_\theta \log \pi_\theta(a_t \mid s_t) \cdot R_{\text{tier}}(\tilde{\tau}) \right]. \tag{14}$$

**Step 3: Baseline Subtraction.** For any $b_t(s_t)$ not depending on $a_t$,

$$\mathbb{E}_{\tilde{\tau}\sim p_\theta}\left[\sum_{t=0}^{T-1}\nabla_\theta\log\pi_\theta(a_t\mid s_t)\cdot b_t(s_t)\right] \tag{15}$$

$$=\sum_{t=0}^{T-1}\mathbb{E}_{s_t}\left[b_t(s_t)\cdot\mathbb{E}_{a_t\sim\pi_\theta(\cdot\mid s_t)}\left[\nabla_\theta\log\pi_\theta(a_t\mid s_t)\right]\right]=0, \tag{16}$$

so

$$\mathbb{E}_{\tilde{\tau}\sim p_\theta}\left[\sum_{t=0}^{T-1}\nabla_\theta\log\pi_\theta(a_t\mid s_t)\cdot(R_{\text{tier}}(\tilde{\tau})-b_t(s_t))\right]=\nabla_\theta J(\theta). \tag{17}$$

$\square$

**Remark 2** (If the reasoner is $\theta$-dependent). *If, instead, $p(\xi\mid\tau)$ depends on $\theta$ (e.g., shared trunk), then*

$$\nabla_\theta\log p_\theta(\tilde{\tau})=\sum_t\nabla_\theta\log\pi_\theta(a_t\mid s_t)\ +\ \nabla_\theta\log p_\theta(\xi\mid\tau),$$

*and an unbiased estimator must add the extra score term $\nabla_\theta\log p_\theta(\xi\mid\tau)$ multiplied by the same return. Alternatively, one may stop gradients through the reasoner or generate $\xi$ using frozen modules to enforce $\theta$-independence.*

**Proposition 3** (Unbiased gradient with $\theta$-dependent reasoner). *If $p_\theta(\xi\mid\tau)$ depends on $\theta$, then*

$$\nabla_\theta J(\theta)=\mathbb{E}\left[\left(\sum_t\nabla_\theta\log\pi_\theta(a_t\mid s_t)+\nabla_\theta\log p_\theta(\xi\mid\tau)\right)R_{tier}(\tilde{\tau})\right],$$

*so the unbiased score-function estimator must include both terms (each may use an appropriate baseline that is independent of the respective sampled variable).*

**Corollary 4.** *Under the $\theta$-independence of $p(\xi\mid\tau)$, the three-tier reward preserves the unbiasedness of the REINFORCE estimator for $\nabla_\theta J(\theta)$. Algorithms such as PPO, GRPO, and BNPO, which optimize clipped or normalized surrogate objectives, remain applicable with this reward; however, their gradient estimates are generally biased (by design) and converge to stationary points of their respective surrogate objectives rather than guaranteeing an unbiased gradient of $J(\theta)$.*

## C    Datasets and Baselines

We evaluate on seven benchmarks that span diverse visual contexts beyond symbolic mathematics: natural images, statistical charts, scientific diagrams, tables, real-world scenes, puzzles, and pattern recognition tasks. This diversity ensures our evaluation captures a broad range of visual reasoning challenges.

- **MathVista** (testmini) (Lu et al., 2024a): A consolidated benchmark of 1000 examples, covering figure QA, geometry, math word problems, and textbook QA across diverse reasoning types.
- **MathVision** (Wang et al., 2024): 3040 problems sourced from real math competitions, spanning 16 mathematical disciplines across 5 difficulty levels.
- **MathVerse** (Zhang et al., 2024): 2612 visual math problems transformed into 6 versions with varying visual vs. textual information to test genuine diagram comprehension.
- **MMMU** (dev & validation) (Yue et al., 2024): 1050 college-level questions across 6 disciplines and 30 image types including charts, diagrams, tables, and scientific figures.
- **WeMath** (testmini, strict) (Qiao et al., 2024): 1740 visual math problems organized around 67 hierarchical knowledge concepts, decomposing composite problems into sub-problems to assess foundational knowledge.
- **DynaMath** (worst-case) (Zou et al., 2025): 501 seed questions represented as Python programs generating variants with different numerical values or geometric transformations to test reasoning robustness.
- **LogicVista** (Xiao et al., 2024): 448 questions evaluating logical reasoning across 5 types (deductive, inductive, spatial, numerical, mechanical) sourced from human IQ and reasoning test banks.

**Model Baselines.**    We compare against leading proprietary models: GPT-4o (Achiam et al., 2023), Claude-3.7-Sonnet, Gemini-2.0-Flash (Team et al., 2023), o1 (Jaech et al., 2024), Gemini 2.5 Pro (Comanici et al., 2025), and Seed1.5-VL (Guo et al., 2025b); open-weights general-purpose models: Qwen2.5-VL (Bai et al., 2025) and Ovis2 (Lu et al., 2024b); and reasoning-optimized models: InternVL3-MPO variants (Zhu et al., 2025), VL-Rethinker (Wang et al., 2025a), QVQ-72B-Preview (Team, 2024b), and MMR1-Math (Sicong Leng, 2025).

## D    Generation Diversity During Training

An interesting emergent property of AC-RL training is that it maintains greater generation diversity compared to standard binary-reward RL. Figure 4 tracks the fraction of training batches where all $M$ generated captions receive identical rewards (zero standard deviation), which serves as an indicator of diversity collapse.

Both methods show an upward trend as entropy naturally decreases during policy optimization. However, AC-RL consistently maintains a lower fraction of uniform-reward batches (approximately 0.31 vs 0.42 at convergence). This difference likely stems from the tiered reward structure: while standard RL only distinguishes between success and failure, AC-RL's intermediate reward ($\alpha = 0.7$) creates a richer gradient landscape that encourages the model to explore different captioning strategies.

## E    Training Dynamics

Figure 5 tracks the (smoothed) frequency of clarification requests during training. The consistent decrease confirms our hypothesis that AC-RL teaches the captioner to front-load information that would otherwise trigger follow-up questions. Combined with the maintained generation diversity shown in Appendix D, this suggests the model finds genuinely informative captioning strategies rather than mode-collapsing to a narrow set of templates.

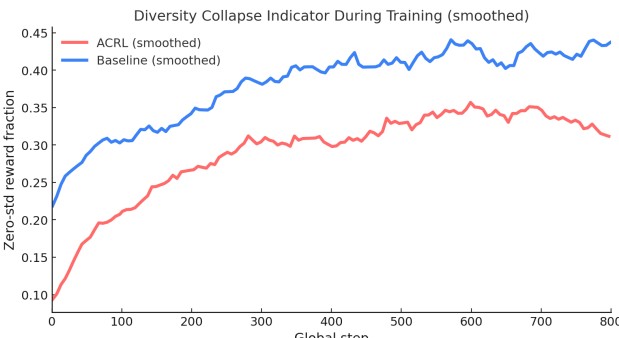

Figure 4: Fraction of uniform-reward batches during training. AC-RL (red) maintains lower values than standard RL (blue), indicating more diverse caption generation throughout training. Both methods show increasing trends as policies converge, but AC-RL's tiered reward structure preserves more exploration.

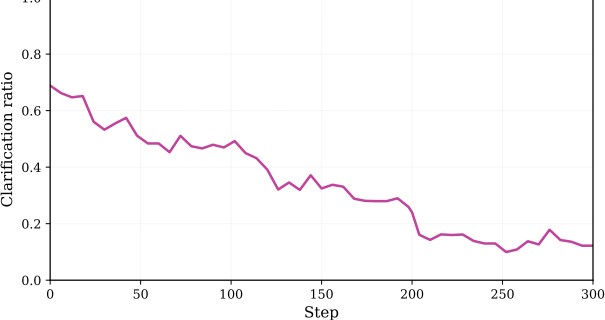

Figure 5: Clarification request frequency during AC-RL training. The rate decreases from approximately 65% at initialization to 20% at convergence, demonstrating that the model learns to produce more self-sufficient initial captions.

## F    Qualitative Caption Comparisons

To illustrate how AC-RL modifies captioning behavior, we present a representative example comparing captions from the pretrained baseline and AC-RL on a table-reading problem (Figure 6).

| | |
|---|---|
| oval beads | $2/kilogram |
| rectangular beads | $3/kilogram |
| star-shaped beads | $2/kilogram |
| spherical beads | $2/kilogram |
| heart-shaped beads | $3/kilogram |
| square beads | $2/kilogram |
| flower-shaped beads | $2/kilogram |

> **Question:** Chase wants to buy 4 kilograms of oval beads and 5 kilograms of star-shaped beads. How much will he spend? (Unit: $)

**Baseline Caption**

The image appears to be a table listing various types of beads along with their respective prices per kilogram. The table has two columns and lists different types of beads available for purchase. I can see oval beads, rectangular beads, star-shaped beads, spherical beads, heart-shaped beads, square beads, and flower-shaped beads. Each type has a corresponding price per kilogram shown in the second column. The table appears to be from a craft supply store or similar retailer.

**AC-RL Caption**

The image appears to be a table listing various types of beads along with their respective prices per kilogram. The table is as follows:

| Beads | Price/kg |
|---|---|
| oval beads | $2 |
| rectangular beads | $3 |
| star-shaped beads | $2 |
| spherical beads | $2 |
| heart-shaped beads | $3 |
| square beads | $2 |
| flower-shaped beads | $2 |

*(Model outputs Markdown; rendered for readability.)*

Figure 6: Qualitative comparison on a table-reading problem. The baseline describes the table structure but omits numerical values. AC-RL extracts all prices needed to compute the answer.

The baseline caption correctly identifies the table structure and all bead types, but refers to prices abstractly without extracting the actual values. AC-RL, having learned from clarification requests during training that numerical details are essential, extracts every price explicitly.

## G    Prompt Templates

We present the complete set of prompts used in our AC-RL framework. The prompts are structured to maintain clear role separation between the captioner (visual description) and reasoner (problem-solving), while enabling controlled interaction during training.

### G.1    Vision-Language Model Prompts

**Initial Caption Generation.**    The following prompt instructs the VLM to generate comprehensive visual descriptions without solving the problem:

```
vlm_initial_description_prompt

I need your help analyzing this image to prepare for answering the following
question:
{question}
IMPORTANT: DO NOT answer the question directly. Instead, provide a
comprehensive and detailed description of everything visible in the image
that could be relevant for answering this question.
Focus on describing:

    • All objects, people, text, and visual elements in the image

    • Spatial relationships between different elements

    • Any text content that is visible, transcribed exactly

    • Colors, shapes, patterns, and visual attributes

    • Relevant contextual details and background information

Your description should be detailed enough that someone could mentally
reconstruct the image without seeing it, but DO NOT provide step-by-step
instructions on how to recreate it.
```

**Clarification Response.** When the reasoner requests specific visual information, the frozen reference model uses this prompt:

```
vlm_focused_description_prompt

Original Question: {question}
Previous Description: {previous_descriptions}
CONTEXT: The description above was provided for this image, but some details
might be missing or unclear. We are asking this specific follow-up question
to gather additional visual details.
Your specific task: {focus_request}
CRITICAL INSTRUCTIONS:

    • You are a VISUAL DESCRIBER only - DO NOT attempt to answer the
      original question

    • DO NOT solve the problem or provide calculations

    • DO NOT give step-by-step solutions or reasoning

    • ONLY describe what you can see in the image that relates to the
      specific request

    • Focus solely on visual elements: objects, text, numbers, shapes,
      spatial relationships

    • If asked about measurements, describe what you see but don't
      calculate or solve

    • If asked about equations, transcribe what's visible but don't solve
      them

    • Be thorough and precise in your description since this is to clarify
      specific missing details
```

## G.2 REASONER PROMPTS

**Adaptive Decision Mechanism.** The reasoner evaluates whether the initial caption is sufficient or requires clarification:

```
reasoner_adaptive_decision_prompt

You are an expert visual reasoning assistant. Your task is to analyze the
given image description and decide if you can solve the problem directly or
if you need one specific piece of additional visual information.
Image Description: {description}
Question: {question}
ANALYSIS INSTRUCTIONS:

    1. CAREFUL EVALUATION: Analyze if the description contains all specific
       visual details needed to solve completely and accurately.

    2. BE CONSERVATIVE: If missing ANY crucial visual detail, request MORE
       information rather than guess.

    3. ONE CLARIFICATION ONLY: You can request specific additional visual
       information if needed.

    4. DECISION CRITERIA:

        • If you have ALL visual details needed: Status = SOLVED
        • If missing crucial visual information: Status = NEED_MORE_INFO

    5. AVOID ASSUMPTIONS: Don't guess numbers, assume "typical" values, or
       fill in missing details.

CRITICAL PRINCIPLES:

    • BE SPECIFIC in requests: Ask for exact details you need

    • SOLVE CONFIDENTLY when possible: If you have enough information,
      provide the complete solution

    • REQUEST STRATEGICALLY: Make your one request count – ask for the most
      crucial missing details

OUTPUT FORMAT (all fields required):
Reasoning: [Your detailed analysis of what information you have and what
might be missing]
Status: [SOLVED or NEED_MORE_INFO]
Answer: [Your complete final answer if Status is SOLVED - use \boxed{answer}
format, otherwise N/A]
Request: [Your specific request for additional visual information if Status
is NEED_MORE_INFO, otherwise N/A]
```

**Final Answer Generation.** For both direct solving and post-clarification scenarios:

```
reasoner_final_prompt

You are an expert mathematical reasoning assistant. Based on the complete
image description below, please solve the mathematical problem step-by-step.
Complete Image Description: {description}
Question: {question}
INSTRUCTIONS:

    1. Analyze the complete image description carefully

    2. Work through the problem step-by-step with clear mathematical
       reasoning

    3. Show all calculations and logical steps

    4. Provide your final answer in the required format

    5. Use \boxed{answer} notation. For multiple choice, use \boxed{letter}
       format
You MUST follow this format:
<think>
Your detailed reasoning and thought process here...
</think>
<answer> Final Answer: your final answer here </answer>
```

## H  TRAINING HYPERPARAMETERS

We provide complete hyperparameter specifications to ensure reproducibility. All experiments use the same random seed for dataset sampling to enable fair comparisons.

Table 8: Hyperparameter settings for AC-RL training across different model sizes.

| Hyperparameter | 2B Models | 3B Models |
|---|---|---|
| *Optimization* | | |
| Learning rate | $3 \times 10^{-6}$ | $2 \times 10^{-6}$ |
| Effective batch size | 256 | 256 |
| KL divergence weight ($\beta$) | 0.001 | 0.001 |
| *LoRA Configuration* | | |
| LoRA rank ($r$) | 128 | 256 |
| LoRA alpha ($\alpha$) | 256 | 512 |
| LoRA dropout | 0.05 | 0.05 |
| *BNPO Settings* | | |
| Group size ($M$) | 8 | 8 |
| Number of iterations | 6 | 6 |
| Remaining parameters | TRL library defaults | |
| *Generation Parameters* | | |
| Captioner temperature | 1.0 | |
| Captioner max tokens | 800 | |
| Reasoner temperature | 0.6 | |
| Reasoner top-$p$ | 0.95 | |
| Reasoner max tokens | 100,000 | |
| *Reward Configuration* | | |
| Clarification penalty ($1 - \alpha$) | 0.3 | |

**Implementation Details.** We implement AC-RL using the Transformers Reinforcement Learning (TRL) library von Werra et al. (2020) with Beta-Normalization Policy Optimiza-

tion (BNPO). The LoRA Hu et al. adapters are applied to all linear layers in the vision-language models. Training typically converges within 1,000 steps on the ViRL-39K dataset. All experiments use mixed precision training (fp16).

**Computational Requirements.** Training a 3B parameter captioner with AC-RL requires approximately 50 hours on 8 NVIDIA A6000-Ada GPUs. The 2B models require 40 hours on the same hardware configuration. The reasoner is hosted on a $4\times$ AMD MI250X node, though it is never fully saturated during training.

# I LLM USAGE STATEMENT

We used large language models for grammatical corrections and rewording suggestions to improve clarity. All research ideas, experimental design, analysis, and scientific contributions are the original work of the authors. LLMs were not used for generating research content or results interpretation.

