# OpenReview forum: "Clarification as Supervision: Reinforcement Learning for Vision-Language Interfaces"
_ICLR.cc/2026/Conference — Submitted to ICLR 2026_

### Official Review · Reviewer_JaUc · 2025-11-01

**Soundness:** 2
**Presentation:** 2
**Contribution:** 2
**Rating:** 4
**Confidence:** 4

**Summary:**

This paper addresses the "interface mismatch" in decoupled vision-reasoning systems, where VLM-generated captions often lack the specific details required by downstream text-only reasoners. The authors propose Adaptive-Clarification Reinforcement Learning (AC-RL), which uses a "training scaffold" where a reasoner can request clarification. The core idea is a tiered reward structure. The authors claim this reward penalty creates optimization pressure for the VLM captioner to "front-load" critical information into its initial description.

**Strengths:**

1. The paper correctly identifies the potential of introducing dense reward and clarification in the reasoning process.

2. The paper include analysis to support how the proposed mechanism works.

**Weaknesses:**

1. Unjustified Premise and Limited Scope: The paper's core premise—that multimodal reasoning is best solved by decoupling perception (a captioner) from reasoning (a text-only LLM)—is presented without justification. This is a counter-intuitive paradigm and not widely adopted as for recent state-of-the-art VLMs, as it assumes complex visual problems can be 'flattened' into a single text pass. This lack of justification is compounded by the paper's narrow benchmark focus. The success on math-heavy tasks suggests this method primarily optimizes visual data extraction (of text, numbers, facts) rather than holistic perceptual reasoning. It is thus narrowing the generalizability of the proposed approach and is highly questionable if this approach would work on perception-intensive tasks (e.g., visual search, complex spatial relations) where iterative grounding with the image is required.

2. The current manuscript suffers from presentation and clarity issues, requiring the reader to make guesses when reading.

- Undefined Jargon: The paper uses terms like "GRPO-style optimization" (Section 4.1) without definition or citation. Other core concepts, like "BNPO" (Section 3.4), are cited but without even an intuitive explanation, forcing the reader to consult external papers to understand the methodology.

- Inferential Leaps: The analysis makes claims that are not fully supported by the data. For instance, in Section 4.3, the data in Table 5 shows that clarification requests from the AC-RL model are more critical; the paper leaps to the conclusion that this proves the model "learns to distinguish between recoverable and irrecoverable information gaps," which is a strong claim for which other explanations exist (e.g., it simply fails to front-load the most difficult facts, which are by definition critical).

- Overstated Conclusions: The conclusion generalizes the findings from math benchmarks to "any modular architecture," which is a significant overstatement of what has been demonstrated.

3. Missing Ablation on Key Hyperparameter ($\alpha$): The partial reward value $\alpha=0.7$ is central to the entire method, but it is presented without any justification or sensitivity analysis. How was this value chosen? How does performance vary with a stronger penalty (e.g., $\alpha=0.3$) or a weaker one ($\alpha=0.9$)? The lack of an ablation study on this critical hyperparameter makes the results difficult to interpret.

4. Risk of Overfitting to the Reasoner: The captioner is trained to adapt to the information needs of a single, frozen reasoner (DeepSeek-R1-Qwen-3B). It is an open question whether the learned captioning policy is generically better or simply "overfit" to the specific quirks and biases of this one reasoner. The claims would be much stronger if the authors showed that the AC-RL-trained captioner also improves performance when paired with a different, unseen reasoner at inference time.

**Questions:**

1. Can you please justify the choice of a decoupled paradigm over a standard end-to-end VLM? Given the counter-intuitive nature of this approach, what evidence suggests it is a necessary or superior path for multimodal reasoning?

2. Could you please define terms like "GRPO-style optimization" and briefly explain the mechanics of "BNPO" as they relate to your method?

3. Could you please provide a sensitivity analysis or ablation study for the clarification reward hyperparameter $\alpha$? How does performance change as this value is varied?

4. Can you elaborate on the "negative clarification gap" in Table 4? Why exactly does allowing clarification hurt the AC-RL model on MathVerse? Does this suggest the frozen clarification model ($\pi_{ref}$) is providing low-quality or conflicting information, and if so, how does that affect the training signal?

5. Related to the WeMath regression (Table 1) and the subject-level analysis (Figure 2), do you see evidence of a trade-off? Does the optimization pressure to "front-load" quantitative and geometric details (where AC-RL excels) come at the cost of performance on other types of reasoning (e.g., logic, descriptive)?

6. How well does the AC-RL-trained captioner generalize? If you were to swap the DeepSeek-based reasoner for a different frozen reasoner (e.g., one based on Llama 3 or Claude 3) at inference time, does the improved captioning policy still provide a performance benefit?

7. Given that the decoupled method excels on tasks requiring visual data extraction (geometry, algebra), how do you expect this approach would perform on benchmarks that are more perception-intensive (e.g., visual search, fine-grained attribute recognition, or abstract spatial reasoning) which may not be solvable by a single, 'front-loaded' text description?

---

> ### Author Response · Authors · 2025-11-26
> **Response to Reviewer JaUc (1/3)**
>
> We appreciate the thorough analysis and feedback about our presentation. We've made substantial revisions to address concerns, and we hope our clarifications help position the contribution more clearly.
>
> ---
>
> > **Weakness 1:** "The paper's core premise—that multimodal reasoning is best solved by decoupling perception (a captioner) from reasoning (a text-only LLM)—is presented without justification. This is a counter-intuitive paradigm..."
>
> > **Question 1:** "Can you please justify the choice of a decoupled paradigm over a standard end-to-end VLM?"
>
> We did not intend to make any broad claim that decoupled architectures are superior to end-to-end VLMs, and adjusted our motivation in the paper.
>
> Our focus has been a specific practical scenario that reflects a common deployment scenario: organizations often have access to strong text-only reasoners (e.g., GPT-5, Claude) via APIs that cannot be fine-tuned, or models that are too large or costly to adapt (e.g., DeepSeek R1 671B). In such cases, the vision front-end is the only component that can realistically be adapted.
>
> Moreover, many domains already rely on specialized vision-language models (e.g., for medical imaging, web interfaces, or engineering diagrams), which organizations may further specialize on their own data. These models, however, typically lack broad reasoning capabilities, making it valuable to pair them with powerful frozen/API-based reasoners. This collaboration pattern (between a specialized local model and a general external agent/solver) is increasingly common in modern agentic and LLM-based systems.
>
> We have clarified this in the introduction (changes in blue), emphasizing the practical constraints that motivate our setup. In sum, we agree that the preference for a modular approach is not inherent but we believe that this setting we consider is common and important.
>
> ---
>
> > **Weakness 1:** "The success on math-heavy tasks suggests this method primarily optimizes visual data extraction ... rather than holistic perceptual reasoning. It is thus narrowing the generalizability... highly questionable if this approach would work on perception-intensive tasks..."
>
> > **Question 7:** "how do you expect this approach would perform on benchmarks that are more perception-intensive (e.g., visual search...)"
>
> We intentionally chose these reasoning benchmarks as a controlled testbed for two reasons: 1) clear ground truth enables clean measurement of interface improvements, 2) strong text-only reasoners exist, isolating the vision-language interface as the bottleneck, 3) easier to train as rewards are verifiable.
>
> We want to emphasize that the reasoning benchmarks that we used are quite visually diverse and cover a broad range of cases. MathVista includes natural images, MathVision statistical charts and puzzles. MMMU covers many scientific disciples requiring the model to interpret and describe tables, scientific diagrams and real-world scenes. LogicVista focuses on puzzle-like problems, pattern recognition and spatial arrangements.
>
> For tasks requiring iterative visual grounding (fine-grained recognition, complex spatial reasoning), the natural extension is multi-round AC-RL where multiple clarification rounds allow progressive refinement. For region-level tasks, clarifications could request descriptions of specific image areas. Whether the core mechanism transfers effectively to these settings is an empirical question that warrants future investigation. In some domains, transitioning to a strict no-clarification regime at test time may be too restrictive. In such cases, an extension of AC-RL could instead target minimal-interaction behaviour, learning effective multi-turn dialogues while still reducing reliance on clarification. These directions remain speculative and warrant deeper investigation.
>
> We've revised Section 1 (final paragraph, in blue) and the conclusion (second paragraph, in blue) to position math-centric reasoning as a deliberate testbed with natural extensions to other domains.

---

> ### Author Response · Authors · 2025-11-26
> **Response to Reviewer JaUc (2/3)**
>
> > **Weakness (Undefined Jargon):** "The paper uses terms like 'GRPO-style optimization'... 'BNPO' (Section 3.4), are cited but without even an intuitive explanation..."
>
> > **Weakness (Overstated Conclusions):** "The conclusion generalizes the findings from math benchmarks to 'any modular architecture,' which is a significant overstatement..."
>
> We appreciate these comments and agree that these were omissions on our part. Subsequently, we have made the following revisions:
>
> **Undefined jargon:** We've added clear definitions at first use. Examples: "GRPO (Group Relative Policy Optimization) is an on-policy RL algorithm that optimize over groups of responses per prompt" (Section 3.4, footnote 1, in blue) and "BNPO (Beta-Normalization Policy Optimization) fits a Beta distribution to the reward distribution for more stable advantage estimation" (Section 3.4, Appendix A).
>
> **Overstated claims:** We've revised the conclusion to scope claims appropriately (Conclusion, paragraph 2, in blue), and we clearly separate demonstrated results from speculative extensions.
>
> All changes are highlighted in blue in the revised manuscript.
>
> ---
>
> > **Weakness 3:** "The partial reward value $\alpha=0.7$ is central to the entire method, but it is presented without any justification or sensitivity analysis. ... The lack of an ablation study on this critical hyperparameter makes the results difficult to interpret."
>
> We've completed a study across $(1 - \alpha) \in \{0.1, 0.3, 0.5, 0.7\}$. To ensure a fair ceteris paribus comparison, we retrained all $\alpha$ configurations with identical hyperparameters and doubled the batch size for training stability. This means our reported penalty=0.3 model achieves 55.70% on LogicVista compared to 53.02% in the main paper. This demonstrates that AC-RL with penalty=0.3 is robust across training conditions and even improves with more stable optimization. The relative comparison across penalty values remains valid and clearly shows penalty=0.3 is optimal.
>
> | Penalty $(1-\alpha)$ | LogicVista | MathVision |
> |:---:|:---:|:---:|
> | 0.1 | 53.24% | 35.79% |
> | **0.3** | **55.70%** | **40.23%** |
> | 0.5 | 52.35% | 40.10% |
> | 0.7 | 50.11% | 35.53% |
>
> The results clearly show penalty=0.3 is optimal. If the penalty is too small (0.1), there's not enough pressure to avoid clarification. If it's too large (0.7), we lose the benefit of partial rewards for clarified successes and essentially revert to binary rewards. We've added this as Table 3 in Section 4.2 (following the Binary-Reward RL ablation, highlighted in blue).
>
> ---
>
> > **Question 4**: "Can you elaborate on the 'negative clarification gap' in Table 4? Why exactly does allowing clarification hurt the AC-RL model on MathVerse? Does this suggest the frozen clarification model (π_ref) is providing low-quality or conflicting information, and if so, how does that affect the training signal?"
>
> We acknowledge that this counterintuitive result deserves attention. Our view is that AC-RL learns to front-load key information into the initial captions for MathVerse problems. In our clarification-enabled evaluation protocol, we prompt the reasoner to ask for clarification when possible (mirroring the training setup). When the initial caption already contains what is needed, forcing an extra clarification can occasionally introduce noise or lead the reasoner into a less optimal reasoning path.
>
> We do not think this is a serious issue during training. First, the reward is only given when the problem is correctly solved, so a weaker clarification run may not introduce much additional noise, though it may slightly reduce reward density. Second, early in training the initial captions tend to be not informative enough, so successful clarification rounds provide valuable signal: a good caption enables a good clarification, and therefore gets reinforced. Later in training, clarification becomes less crucial / informative (and less dense than for other datasets). This interpretation aligns with the training dynamics plot (Appendix E, Figure 5), where the clarification rate drops from ~65% to ~20% over the course of training.
>
> Note that under single-pass deployment which is our target setting, this issue does not arise, and AC-RL consistently outperforms both the pretrained baseline and binary-reward RL.

---

> > ### Author Response · Authors · 2025-11-26
> > **Response to Reviewer JaUc (3/3)**
> >
> > > **Weakness 4:** "Risk of Overfitting to the Reasoner: ... It is an open question whether the learned captioning policy is generically better or simply 'overfit' to the specific quirks and biases of this one reasoner."
> >
> > > **Question 6:** "If you were to swap the DeepSeek-based reasoner for a different frozen reasoner ... at inference time, does the improved captioning policy still provide a performance benefit?"
> >
> > We've begun cross-reasoner evaluation to assess whether AC-RL-trained captioners generalize beyond the training reasoner. We trained with DeepSeek-R1-Qwen-32B and are testing with an alternative reasoner at inference:
> >
> > **Preliminary Results (LogicVista)**
> > - Baseline captioner + DeepSeek-R1-0528-Qwen3-8B: 44.29%
> > - Baseline captioner + DeepSeek-R1-Qwen-32B (training reasoner): 47.2%
> > - AC-RL captioner + DeepSeek-R1-0528-Qwen3-8B: 53.02%
> > - AC-RL captioner + DeepSeek-R1-Qwen-32B (training reasoner): 55.70%
> >
> > **Preliminary Results (MathVision)**
> > - Baseline captioner + DeepSeek-R1-0528-Qwen3-8B: 30.0%
> > - Baseline captioner + DeepSeek-R1-Qwen-32B (training reasoner): 34.7%
> > - AC-RL captioner + DeepSeek-R1-0528-Qwen3-8B: 36.51%
> > - AC-RL captioner + DeepSeek-R1-Qwen-32B (training reasoner): 40.23%
> >
> > The results suggest AC-RL-trained captioners maintain strong performance across reasoners, though some performance variation is expected since we optimize for a specific reasoner during training. That being said, we are not sure that overfitting is necessarily such a big problem; in practice, if there is a need to use a different reasoner at test time, extra fine-tuning of the captioner can be performed.
> >
> > We've added this preliminary analysis to Section 4.2 (new subsection after $\alpha$ ablation, highlighted in blue).
> >
> > ---
> >
> > > **Question:** "Related to the WeMath regression (Table 1) and the subject-level analysis (Figure 2), do you see evidence of a trade-off? Does the optimization pressure to 'front-load' ... come at the cost of performance on other types of reasoning?"
> >
> > The modest WeMath regressions (-0.7 to -1.5 points) are worth examining. We posit that the problems emphasize different aspects of reasoning where our visual interface improvements are less critical.
> >
> > More specifically, we hypothesize that AC-RL's optimization pressure prioritizes information extraction (mining the image for missing details), which yields massive gains on extraction-heavy benchmarks like DynaMath (+10.6%). On the other hand, WeMath is designed to evaluate human-like reasoning based on preexisting knowledge. It appears that knowledge extraction from the image is not the main bottleneck here: even without additional training, the base model already identifies relevant visual content fairly well. The core difficulty in this benchmark instead lies in reasoning: failures typically occur because the model lacks the required theorem or formula, rather than because it misunderstands the image and AC-RL has no effect on reasoning. This does not fully explain the slight drop in performance, but it makes the lack of improvement more understandable. We plan to include a more systematic discussion of this in the next revision.
> >
> > Subject-level analysis (Figure 2) shows AC-RL specifically and substantially improves quantitatively-intensive domains (geometry +10.4 points, algebra +15.1 points) while remaining roughly neutral elsewhere. This suggests targeted improvement where visual information matters most.
> >
> > We've added a discussion to Section 4.1 noting the WeMath pattern.
> >
> > ---
> >
> > ### What We've Changed
> >
> > We've addressed all major concerns with these revisions:
> >
> > - **Section 1, paragraphs 2-3:** Explicit practical motivation without universal architectural claims
> > - **Section 3.1, paragraph 1:** Framework positioned for practical deployment scenario
> > - **Section 3.4:** GRPO footnote and BNPO definition with explanations
> > - **Ablations subsection:** New table with $\alpha$ ablation
> > - **Ablations subsection:** New subsection on preliminary cross-reasoner generalization results
> > - **Section 4.1:** Discussion of WeMath performance patterns
> > - **Conclusion, paragraph 2:** Revised scope, separated demonstrated vs. speculative results
> >
> > Given these substantial revisions and additional results, we hope you'll reconsider your assessment.

---

### Official Review · Reviewer_cc99 · 2025-11-01

**Soundness:** 3
**Presentation:** 3
**Contribution:** 3
**Rating:** 6
**Confidence:** 4

**Summary:**

This paper introduces Adaptive-Clarification Reinforcement Learning (AC-RL), a framework that teaches vision-language models to generate reasoner-aligned captions by treating clarification requests as implicit supervision. The model uses a tiered reward structure that penalizes reliance on clarifications during training, encouraging more information-rich initial captions for single-pass inference. Empirical results on seven math reasoning benchmarks show that AC-RL achieves consistent gains over several baselines.

**Strengths:**

1. The approach of using clarification as supervision through multi-turn interaction is novel and well-motivated.
2. AC-RL consistently outperforms baseline methods (binary-reward RL and decoupled architectures) under single-pass inference on extensive mathematical reasoning benchmarks, confirming effectiveness without adding inference cost.

**Weaknesses:**

[Major Weakness]
1. Tab. 4 shows AC-RL performs worse under clarification-enabled evaluation on MathVerse_MINI, compared to single-pass (34.26% vs 36.80%), which is counter-intuitive since more interactions should help, but the existing provided justification (i.e., additional clarification introduces noise to well-aligned caption) is not convincing enough, given the slight improvement on MathVision and the fact that the reasoner knows to stop requesting once it is satisfied.
2. The qualitative comparison of the generated captions from the captioner before and after RL is missing, which helps to illustrate how AC-RL improves beyond binary reward RL.
3. Training logs that track the frequency of clarification requests from the reasoner (e.g., plotted over every 100 steps) should be provided to show how the policy adapts over time.

[Minor Weakness]
1. Additional analyses are potentially insightful, and thus recommended to provide to benefit the community:
     - A. Reporting clarification-enabled performance on additional benchmarks (e.g., DynaMath and LogicVista), along with a cost/benefit analysis of extra interaction rounds, would provide a more complete evaluation.
     - B. Plotting accuracy as a function of the number of clarification rounds (and observing whether the curve saturates) would help clarify how much clarification is actually beneficial.
2. The definition of $Acc_{deny}$ in Tab. 5 and the calculation process of obtaining it is unclear, which should be clearly stated.
3. Tab. 1 lists "Qwen-3B" which is a text-reasoning model, not a VLM, making it unclear how visual reasoning results are obtained for this configuration.

**Questions:**

1. Will the codebase be made public to the community?

---

> ### Author Response · Authors · 2025-11-26
> **Response to Reviewer cc99**
>
> Thank you for the positive assessment and for recognizing that our approach is "novel and well-motivated." We appreciate the detailed suggestions for additional analyses. We've completed the requested experiments and believe they significantly strengthen the empirical validation.
>
> > **Major Weakness 1:** "Tab. 4 shows AC-RL performs worse under clarification-enabled evaluation on MathVerse_MINI... which is counter-intuitive since more interactions should help, but the existing provided justification ... is not convincing enough..."
>
> We acknowledge this counterintuitive result deserves careful interpretation. Our explanation is that AC-RL has learned to front-load critical information into initial captions for MathVerse problems. In our clarification-enabled evaluation protocol, we prompt the reasoner to attempt clarification when possible (following the training setup). When the initial caption is already informative, forcing an additional clarification might occasionally introduce noise or lead the reasoner down different reasoning paths.
>
> However, we emphasize this is speculation based on the observed pattern. The key practical point is that in real single-pass inference, where clarification isn't available, AC-RL achieves 36.80% accuracy, which is the highest performance. The negative gap under clarification-enabled evaluation doesn't affect the deployment scenario our method targets.
>
> ---
>
> > **Major Weakness 2:** "The qualitative comparison of the generated captions from the captioner before and after RL is missing, which helps to illustrate how AC-RL improves beyond binary reward RL."
>
> We've prepared detailed side-by-side comparisons showing how AC-RL improves captions compared to the baseline model. ACRL provides the details required to answer the question, in contrast to the baseline caption.
>
> We've added a section (Appendix F) showing such an example side-by-side with the corresponding analysis.
>
> ---
>
> > **Major Weakness 3:** "Training logs that track the frequency of clarification requests from the reasoner (e.g., plotted over every 100 steps) should be provided to show how the policy adapts over time."
>
> We've generated training logs tracking clarification request frequency. The (smoothed) plot shows clarification frequency starts at ~65% at initialization and decreases to ~20% at convergence, demonstrating the model requests fewer clarifications the further in training it is.
>
> Additionally, AC-RL maintains more varied generations throughout training compared to binary-reward RL (as shown in Appendix D Figure 4). The combination of decreasing clarification rate with maintained diversity suggests the model finds genuinely informative caption strategies rather than mode-collapsing.
>
> We've added this as Figure 5 in Appendix E (before the qualitative examples, highlighted in blue) with accompanying analysis paragraph.
>
> ---
>
> > **Minor Weakness 1A:** "Reporting clarification-enabled performance on additional benchmarks (e.g., DynaMath and LogicVista)... would provide a more complete evaluation."
>
> We are working on this but may not be able to have the results before the end of the rebuttal period, but should have them for a subsequent revision.
>
> ---
>
> ### Minor Clarifications
>
> **Acc_deny definition:** We've clarified Table 7’s caption. Acc_deny is computed on the subset of problems where the reasoner requested clarification during clarification-enabled evaluation. For these specific problems, we measure accuracy by solving the problem using only the initial caption. This isolates the performance impact on problems where the model thinks it needs clarification. (Table 7 caption, revised in blue)
>
> **Qwen-3B naming:** This should be Qwen2.5-VL-3B throughout. We've fixed this typo in all tables and text.
>
> **Code release:** Yes, we will be releasing the complete codebase, training scripts, and model checkpoints upon acceptance.
>
> ---
>
> ### What We've Changed
>
> We've addressed your suggestions with these additions:
>
> - **Appendix E (new):** Training dynamics figure showing clarification frequency over training steps
> - **Appendix F (new):** Side-by-side qualitative caption example
> - **Table 7 caption:** Clarified Acc_deny definition
> - **Throughout tables and text:** Fixed Qwen-3B $\to$ Qwen2.5-VL-3B
>
> We hope that these changes address most of your concerns, but we will be happy to discuss any further questions.

---

### Official Review · Reviewer_5pEc · 2025-11-03

**Soundness:** 2
**Presentation:** 2
**Contribution:** 2
**Rating:** 2
**Confidence:** 3

**Summary:**

This paper presents AC-RL, a method that trains a caption model to generate visual captions that contains information needed for the accompanying reasoning model to answer the question correctly. The core idea of the paper is to assign partial rewards to captions that gets the correct answer by asking additional clarification questions. This will make the reward denser and encourage the captioner to generate more detailed caption with the necessary information needed for the reasoning model. The paper conducts experiments on multi-modal reasoning benchmarks using InternVL2B model and Qwen3B model as the captioning model and the text only deepseek model as the reasoning model and shows that such training enables the reasoning model to better answer the question.

**Strengths:**

- The paper proposes a new method to more effectively train a captioning model to output the information needed for a reasoning model by assigning partial reward to a caption when the model can use the caption to answer the question correctly with clarification.
- The paper demonstrates the effectiveness of their method on multiple math reasoning benchmarks using 2 different models. The performance does show gain over baseline methods.

**Weaknesses:**

- the hyper parameter alpha here is arbitrarily chosen.
- The reward design is problematic, the method is assigning partial rewards to the cases where the reasoning model decides to ask for clarification and then gets the correct answer, and assigns zero reward when the reasoning model gets the incorrect answer no matter what the caption model generates. However, this reward design is not suitable to train the caption model, because when the answer is wrong, it could be that (1) the caption is not good enough (2) the caption is already good but the reasoner simply can't solve the problem since the problem is too hard. Assigning zero could be wrongly penalizing a good caption.
- The reward assignment using clarification is not well motivated. The reasoner model can always chose to ask for clarification when the caption is not detailed enough (eg, an empty string), and in such case, whether the caption gets penalized or rewarded solely depends on the capacity of the reasoner model and the clarification model, not from the behavior of the caption model itself. This does not really make sense.
- The caption model still receives a sparse reward and not any feedback from the clarification process. I think it would make more sense to have the caption model refine their caption based on the clarification process (eg, what questions are asked) since this is an important signal from the reasoner model.

**Questions:**

- How is the hyper-parameter alpha chosen here in this paper?
- Could the author give an explanation/consideration of the design choices? See weakness section.

---

> ### Author Response · Authors · 2025-11-26
> **Response to Reviewer 5pEc (1/2)**
>
> Thank you for the careful reading and detailed questions about our reward structure. We appreciate the opportunity to clarify the design rationale, and also updated the paper extensively to make it much clearer. We discuss your comments below, one by one.
>
> ---
>
> > **Weakness:** "the hyper parameter alpha here is arbitrarily chosen."
>
> > **Question:** "How is the hyper-parameter alpha chosen here in this paper?"
>
> We've completed a study across $(1 - \alpha) \in \{0.1, 0.3, 0.5, 0.7\}$. To ensure a strict ceteris paribus comparison, we retrained all $\alpha$ configurations with identical hyperparameters and doubled the batch size for training stability. This means our reported penalty=0.3 model achieves 55.70% on LogicVista compared to 53.02% in the main paper. This demonstrates that AC-RL with penalty=0.3 is robust across training conditions and improves with more stable optimization. The relative comparison across penalty values remains valid and clearly shows penalty=0.3 is optimal.
>
> | Penalty $(1-\alpha)$ | LogicVista | MathVision |
> |:---:|:---:|:---:|
> | 0.1 | 53.24% | 35.79% |
> | **0.3** | **55.70%** | **40.23%** |
> | 0.5 | 52.35% | 40.10% |
> | 0.7 | 50.11% | 35.53% |
>
> The results clearly show penalty=0.3 is optimal. If the penalty is too small (0.1), there's not enough pressure to avoid clarification. If it's too large (0.7), we lose the benefit of partial rewards for clarified successes and essentially revert to binary rewards. We've added this as Table 3 in Section 4.2 (following the Binary-Reward RL ablation, highlighted in blue).
>
> ---
>
> > **Weakness:** "The reward design is problematic... assigns zero reward when the reasoning model gets the incorrect answer no matter what the caption model generates. ... the caption is already good but the reasoner simply can't solve the problem since the problem is too hard. Assigning zero could be wrongly penalizing a good caption."
>
> We appreciate the reviewer's concern, but we believe there is a misunderstanding of how the reward is used in GRPO.
>
> It is important to remember that training uses multiple rollouts/answers for the same task. When the problem is too hard, the reasoner fails on all rollouts, and all rewards become identical. In this case, with GRPO, the gradient collapses to exactly zero - meaning the caption model receives no update.
>
> In contrast, when at least one caption enables the reasoner to solve the problem, GRPO does produce a gradient: captions that lead to successful reasoning receive positive credit, while less helpful captions are implicitly penalised.
>
> Our method extends this by introducing tiered reward. If no caption directly solves the task, but one caption helps the reasoner ask a useful clarification question and succeed, then that caption is promoted. This is intuitive: a caption that guides the reasoner toward the right question must contain some useful information.
>
> We added the clarification to the paper, see Section 3.4 in blue for the intuitive discussion of learning, and also see Section 3.1 for more intuitive of explanation what drives learning.

---

> > ### Author Response · Authors · 2025-11-26
> > **Response to Reviewer 5pEc (2/2)**
> >
> > > **Weakness:** "The reasoner model can always chose to ask for clarification when the caption is not detailed enough (eg, an empty string), and in such case, whether the caption gets penalized or rewarded solely depends on the capacity of the reasoner model and the clarification model, not from the behavior of the caption model itself."
> >
> > There may be a similar misunderstanding as in the previous point: penalization/credit depends on relative performance between rollouts, not on absolute rewards. Specifically:
> >
> > 1) The reasoner always attempts to solve the task directly from the caption in training. Only when it fails does the clarification stage occur. Therefore, if a caption enables the reasoner to solve the task without clarification while another caption leads to a clarification request, the first caption receives higher credit via the gradient update. This promotes captions that are more informative upfront.
> >
> > 2) When captions are equally unhelpful. If all captions from a given rollout are effectively useless and all solutions come from clarification (i.e., the clarifier and reasoner do all the work), then the rewards - in expectation - are identical across runs. As a result, no gradient update will occur. This is the correct behavior: the model should not arbitrarily prefer one caption when no evidence suggests it is better than the others.
> >
> > We acknowledge that noise may lead to occasional differences between equally good (or equally poor) captions, producing slight reward variations. However, over multiple rollouts this noise cancels out and does not systematically bias training.
> >
> > In summary, the clarification mechanism does not undermine the learning signal. Rather, it allows for partial credit: captions that make the task solvable without clarification are favored; in their absence, captions which carry enough information to ask a good clarification question get a credit. If everything fails, the example is effectively ignored.
> >
> > We've added an explanation to Section 3.1 (new paragraph explaining implicit supervision mechanism, highlighted in blue).
> >
> > ---
> >
> > > **Weakness:** "The caption model still receives a sparse reward and not any feedback from the clarification process. I think it would make more sense to have the caption model refine their caption based on the clarification process (eg, what questions are asked) since this is an important signal..."
> >
> > We thank you for the suggestion. Having the captioner iteratively refine its description based on clarification questions would provide richer supervision than our scalar reward. We agree this is an interesting direction for future work. However, this substantially complicates the training procedure. It requires designing a refinement mechanism, handling multi-turn generation, and correctly propagating the learning signal to each component. Our goal was to show that even a simple scalar three-tier reward structure is sufficient to achieve substantial improvements (+4.4 points on average, up to +5.7 points on LogicVista with our retraining).
> >
> > Importantly, our approach is general and application-agnostic: it applies to any setting where one agent collaborates with a frozen or API-based model. It offers a novel, simple, plug-and-play training method - a lightweight way to train cooperative agents without modifying the underlying models. Our system requires minimal modification to existing RL pipelines and treats the reasoner and clarifier as black boxes. We've added your suggestion about using clarification content more directly to future work (Section 6, paragraph 4, in blue) and appreciate the idea.
> >
> > ---
> >
> > ### What We've Changed
> >
> > Based on your feedback, we've made these revisions:
> >
> > - **Section 3.1, new paragraph:** Additional explanations for learning from clarification patterns
> > - **Section 3.4, new paragraph:** Clarification for "too hard" problem handling
> > - **Ablations subsection:** New table with $\alpha$ ablation
> > - **Section 6, paragraph 4:** Future work on richer clarification-based supervision
> >
> > All changes highlighted in blue.
> >
> > We hope that these clarifications and examples address your concerns about the method's soundness. Given these substantial revisions and clarifications, we hope you'll reconsider your assessment.

---

### Official Review · Reviewer_13NB · 2025-11-03

**Soundness:** 2
**Presentation:** 3
**Contribution:** 2
**Rating:** 4
**Confidence:** 5

**Summary:**

This paper introduces Adaptive-Clarification Reinforcement Learning (AC-RL), a reinforcement learning framework designed to improve the alignment between vision-language models (captioners) and downstream reasoning systems. The key idea is to use clarification requests during training as implicit supervision, encouraging the captioner to generate more comprehensive initial descriptions that reduce the need for multi-turn interaction. The method is evaluated on seven visual mathematical reasoning benchmarks, where it improves average accuracy by up to 4.4 points and reduces clarification dependency by up to 39%.

**Strengths:**

- The tiered reward structure based on clarification need is a clever and intuitive way to address sparse reward challenges in RL. It provides richer training signals and encourages the model to anticipate and preemptively address the reasoner’s information needs.
- The paper includes extensive experiments across seven diverse mathematical reasoning benchmarks, with comparisons to strong proprietary and open-weight models. Ablation studies and behavioral analyses provide strong empirical support for the method’s effectiveness.
- The framework is designed for single-pass inference, making it highly applicable to real-world systems where multi-turn interaction is infeasible. The use of frozen reasoners and reference models during training also ensures modularity and compatibility with existing architectures.

**Weaknesses:**

- The method assumes a frozen, fixed reasoner during training. This limits the framework’s adaptability in scenarios where the reasoner is updated or fine-tuned, which is common in practice. Co-adaptation between the captioner and reasoner is not explored, potentially leaving performance gains on the table.
- The reliance on structured clarification may not transfer well to less formal or more open-ended tasks.
- The choice of the partial reward value (α = 0.7) is not thoroughly justified or ablated. The performance may be sensitive to this hyperparameter, and its optimal value could vary across tasks or reasoners. A sensitivity analysis would strengthen the robustness claim.
- The clarification responses during training are generated by a frozen reference model, which may limit the quality and diversity of the supervision signal.

**Questions:**

Have you experimented with non-mathematical vision-language tasks (e.g., VQA)?

---

> ### Author Response · Authors · 2025-11-26
> **Response to Reviewer 13NB (1/2)**
>
> We thank the reviewer for the detailed feedback and for noting that our tiered reward structure addresses the sparse reward challenge in an intuitive way. We're glad the single-pass inference design resonates as practically applicable. We address your concerns below.
>
> ---
>
> > **Weakness:** "The method assumes a frozen, fixed reasoner during training. This limits the framework's adaptability in scenarios where the reasoner is updated or fine-tuned, which is common in practice. Co-adaptation between the captioner and reasoner is not explored..."
>
> The frozen reasoner constraint is intentional and reflects a common deployment scenario: organizations often have access to strong text-only reasoners (e.g., GPT-5, Claude) via APIs that cannot be fine-tuned, or models that are too large or costly to adapt (e.g., DeepSeek R1 671B). In such cases, the vision front-end is the only component that can realistically be adapted.
>
> Moreover, many domains already rely on specialized vision-language models (e.g., for medical imaging, web interfaces, or engineering diagrams), which organizations may further specialize on their own data. These models, however, typically lack broad reasoning capabilities, making it valuable to pair them with powerful frozen/API-based reasoners. This collaboration pattern (between a specialized local model and a general external agent/solver) is increasingly common in modern agentic and LLM-based systems.
>
> We have clarified this in the introduction, emphasizing the practical constraints that motivate our setup. We acknowledge that in settings where both models can be fine-tuned (or where a multimodal API-based model is effective) alternative strategies may be preferable.
>
> From a technical perspective, this constraint in fact makes our problem harder. The captioner must on its own learn to match a fixed target through pure interaction, with no co-adaptation to make the reasoner more accommodating, only relying on sparse rewards. That said, bidirectional adaptation where both modules co-evolve is an interesting extension. AC-RL can naturally accommodate jointly trainable modules, though this introduces non-stationarity requiring additional technical considerations. We've revised Section 3 (paragraph 1, highlighted in blue) to explicitly frame the frozen reasoner as a practical constraint we're targeting, and added this extension to future work (Section 6, paragraph 3, in blue).
>
> ---
>
> > **Weakness:** "The reliance on structured clarification may not transfer well to less formal or more open-ended tasks."
> > **Question:** "Have you experimented with non-mathematical vision-language tasks (e.g., VQA)?"
>
> AC-RL doesn't hardcode any particular clarification structure. It only requires a) a binary task reward, b) detecting when clarification occurs, and c) a mechanism to provide it during training. The specific form of clarification is domain-dependent and not baked into the algorithm. For example, in open-ended VQA, clarifications might look quite different from our math setting. If the initial caption says "A person is in a room," the reasoner might request "What color is the person's clothing?" or "Describe the objects on the table." The captioner would learn from these patterns to include clothing descriptions and detailed object inventories in future captions. We believe the key mechanism transfers directly across domains.
>
> Importantly, we want to emphasize that the reasoning benchmarks that we used are quite visually diverse and cover a broad range of cases, not only symbolic math, geometry problems and logic puzzles. MathVista includes natural images, MathVision statistical charts and puzzles. MMMU covers many scientific disciplines requiring the model to interpret and describe tables, scientific diagrams and real-world scenes. LogicVista focuses on puzzle-like problems, pattern recognition and spatial arrangements. We adjusted the paper (see in blue) to clarify this.
>
> We intentionally focused on reasoning benchmarks for three reasons. (1) The gap between VLMs and strong reasoning LLMs is especially large on math-like tasks, where post-training with verifiable rewards (e.g., in math and coding) gives text-only reasoners a significant advantage; so the benefits of collaboration are likely most pronounced in this regime; (2) these benchmarks provide clear ground truth and simple metrics, allowing us to isolate the vision–reasoner interaction without confounding factors; (3) extending our approach to training (not only evaluation) on non-math domains would require replacing verifiable rewards with noisier supervision (e.g., LLM-as-a-judge), adding an extra layer of complexity. We've revised Section 4.1 (paragraph 1, in blue) to make this scope choice explicit and added discussion of how the framework extends to other domains in Section 6 (paragraph 2, in blue).

---

> > ### Author Response · Authors · 2025-11-26
> > **Response to Reviewer 13NB (2/2)**
> >
> > > **Weakness:** "The choice of the partial reward value ($\alpha = 0.7$) is not thoroughly justified or ablated. The performance may be sensitive to this hyperparameter... A sensitivity analysis would strengthen the robustness claim."
> >
> > We've completed a study across $(1 - \alpha) \in \{0.1, 0.3, 0.5, 0.7\}$. To ensure a stricter ceteris paribus comparison, we retrained all $\alpha$ configurations with identical hyperparameters and doubled the batch size for training stability. As the result, our reported penalty=0.3 model achieves 55.70% on LogicVista compared to 53.02% in the main paper. This demonstrates that AC-RL with penalty=0.3 is robust across training conditions and improves with more stable optimization. The relative comparison across penalty values remains valid and clearly shows penalty=0.3 is optimal. We've added this as Table 3 in Section 4.2 (following the Binary-Reward RL ablation, highlighted in blue) with accompanying analysis.
> >
> > | Penalty $(1-\alpha)$ | LogicVista | MathVision |
> > |:---:|:---:|:---:|
> > | 0.1 | 53.24% | 35.79% |
> > | **0.3** | **55.70%** | **40.23%** |
> > | 0.5 | 52.35% | 40.10% |
> > | 0.7 | 50.11% | 35.53% |
> >
> > The results show a clear peak at penalty=0.3. Too small a penalty (0.1) doesn't create enough pressure to avoid clarification, while too large (0.7) approaches binary rewards and loses the benefit of densified feedback.
> >
> > ---
> >
> > > **Weakness:** "The clarification responses during training are generated by a frozen reference model, which may limit the quality and diversity of the supervision signal."
> >
> > To address this question, let us discuss two possible alternatives to using the frozen model (please let us know if you had something else in mind):
> >
> > **Option 1:** continuously optimize the captioner for both initial and clarification responses.
> >
> > **Option 2:** optimize the captioner only for the initial response (as in our current setup), and reuse the same model to generate clarifications.
> >
> > We believe Option 1 is problematic: if the clarifier is trainable, the model may achieve high rewards simply by improving the clarification stage, even though clarifications are unavailable at test time. This creates an undesirable failure mode where the clarifier starts providing extra information that should have been included initially. In other words, the captioner may 'hide behind' a strong clarifier rather than learning to generate sufficiently informative initial captions.
> >
> > Option 2 is even more undesirable. If we continuously optimize only for initial captions, the clarifier would gradually become weaker, as the model shifts its focus toward captioning rather than clarification. This could degrade the effectiveness of the supervision signal over time.
> >
> > For these reasons, keeping the clarifier frozen provides us with a stable and reliable supervision process and avoids the above failure modes. We will clarify this motivation in the revised version, and also include option 1 and 2 in an ablation. (An additional practical advantage of our setup is that the clarification step can run on a separate inference-only node. In contrast, options 1 and 2 are more complex - and more costly - within our current infrastructure, hence these results will likely not be available before the end of the rebuttal period.)
> >
> > ---
> >
> > ### What We've Changed
> >
> > We've made substantial revisions to the paper based on your feedback:
> >
> > - **Section 3.1, paragraph 1:** Explicit framing of frozen reasoner as practical constraint and technical challenge
> > - **Section 3.3, paragraph 2:** Detailed frozen clarifier design rationale
> > - **Section 4.1, paragraph 1:** Clarification of scope choice and framework applicability
> > - **Ablations subsection:** New section with $\alpha$ ablation
> > - **Section 6, paragraphs 2-3:** Extensions to other domains and bidirectional adaptation future work
> >
> > All changes highlighted in blue.
> >
> > We hope that these changes address your concerns but we would be happy to provide any extra details.

---

### Author Response · Authors · 2025-12-03
**Common Response (1/2)**

Dear AC,

Thank you for overseeing the review of our paper. Below, we provide a summary of our rebuttal and the reviewer feedback.

The initial scores were (4, 2, 6, 4). We provided detailed responses, conducted additional experiments, and made substantial manuscript revisions addressing all concerns. Reviewer **5pEc** (score 2) raised concerns about reward design that appear to stem from a misunderstanding of GRPO: they worried that zero rewards wrongly penalize good captions on hard problems, but GRPO uses *relative* performance across rollouts, so when all rollouts fail identically, the gradient is exactly zero, and such problems are effectively ignored. We clarified this in our response and revised Section 3.4. The remaining reviewers requested specific additions: **13NB** and **JaUc** asked for $\alpha$ sensitivity analysis and cross-reasoner generalization (now Tables 3-4); **cc99** requested training dynamics and qualitative examples (now Appendices E-F). All requests have been fulfilled.

---

### Strength highlights by the reviewers

- **"Novel and well-motivated"** approach using clarification as supervision through multi-turn interaction. **cc99**
- Tiered reward structure is **"clever and intuitive"** for addressing sparse reward challenges in RL. **13NB**
- **"Extensive experiments"** across seven diverse benchmarks with strong empirical support. **13NB, cc99**
- Framework designed for **single-pass inference**, highly applicable to real-world systems where multi-turn interaction is infeasible. **13NB**
- **"Consistently outperforms baseline methods"** under single-pass inference, confirming effectiveness without adding inference cost. **cc99**
- Includes **analysis to support the mechanism**, demonstrating how the policy adapts over time. **JaUc**

---

> ### Author Response · Authors · 2025-12-03
> **Common Response (2/2)**
>
> ### Concerns raised and our responses
>
> | Category | Core Concern | Response Summary | Manuscript Edits |
> |----------|--------------|------------------|------------------|
> | **1. Fixed Reasoner During Training** | The text reasoner remains frozen during training, which may limit adaptability; co-adaptation is not explored. **13NB, JaUc** | Intentional design reflecting common deployment: API-only reasoners or models too large to fine-tune. Makes the problem harder as the captioner must unilaterally adapt. Despite this constraint, AC-RL improves accuracy by +4.4 points on average, and cross-reasoner experiments show gains transfer to unseen reasoners. Bidirectional adaptation is discussed as future work. | Section 1 (new paragraph on practical motivation), Section 3.1 (paragraph 1), Section 6 (last paragraph) |
> | **2. Hyperparameter $\alpha$ Sensitivity** | $\alpha=0.7$ chosen without justification or ablation. **13NB, 5pEc, JaUc** | Added full ablation across penalties ${0.1, 0.3, 0.5, 0.7}$. Penalty = $1 - \alpha = 0.3$ is optimal; too weak provides insufficient pressure, too strong loses partial credit benefits. | New Table 3, New Section 4.2.2 |
> | **3. Reward Design and Learning Signal** | Zero reward for incorrect answers may wrongly penalize good captions on hard problems. **5pEc** | GRPO uses relative performance across rollouts. When all rollouts fail identically, the gradient is exactly zero, and the problem is effectively ignored. Only captions that perform better than alternatives receive credit. | Section 3.1 (new paragraph 2), Section 3.3 (new paragraph 2), Section 3.4 (new paragraph 2) |
> | **4. Cross-Reasoner Generalization** | Captioner may overfit to the training reasoner's quirks. **JaUc** | Added cross-reasoner evaluation: AC-RL captioner with unseen 8B reasoner achieves +8.7 (LogicVista), +6.5 (MathVision) gains. These improvements are similar to those observed with the original training reasoner, indicating the captioner learns generally informative descriptions rather than overfitting to one reasoner's quirks. | New Table 4, New Section 4.2.3 |
> | **5. Limited Task Scope** | Math-heavy benchmarks; questionable transfer to perception-intensive tasks. **13NB, JaUc** | Benchmarks are visually diverse (natural images, charts, tables, diagrams, puzzles). Math was chosen as a controlled testbed with a clear ground truth. For perception-intensive tasks requiring iterative image queries, extending AC-RL to multiple clarification rounds is a natural direction discussed in future work. | Section 4.1 (new paragraph 1), Section 6 (revised paragraph 2), New Appendix C (dataset descriptions) |
> | **6. Presentation and Clarity** | Undefined jargon (GRPO, BNPO); missing qualitative examples and training dynamics. **JaUc, cc99** | Added definitions with intuitive explanations; added training dynamics figure showing clarification rate decreasing from 65% to 20%; added qualitative caption comparison. | Section 3.4 (new footnote and paragraph), New Appendix A.2 (BNPO explanation), New Appendix E (training dynamics figure), New Appendix F (qualitative examples) |
> | **7. Negative Clarification Gap** | AC-RL performs worse with clarification on MathVerse, which is counterintuitive. **cc99, JaUc** | In an ablation comparing single-pass vs. clarification-enabled evaluation, AC-RL performs slightly worse when clarification is allowed on MathVerse. This is because AC-RL has already learned to include critical details in the initial caption; adding an unnecessary clarification step can introduce noise. Importantly, under single-pass evaluation (our target deployment setting), AC-RL achieves the highest accuracy. | Table 5 caption clarification, Section 4.3 discussion |
>
> ---
>
> ### Summary of All Manuscript Changes
>
> All changes are highlighted in blue in the revised manuscript:
>
> - **Introduction**: Added practical motivation for decoupled architectures (API constraints, specialized VLMs for medical imaging, engineering diagrams)
> - **Section 3.1**: Explicit framing of frozen reasoner as a practical constraint; new explanation of the implicit supervision mechanism
> - **Section 3.3**: Detailed rationale for frozen clarifier design
> - **Section 3.4**: Added GRPO definition, BNPO footnote with explanation, clarification of gradient flow
> - **Section 4.1**: Clarified scope choice and benchmark diversity
> - **Section 4.2**: Two new subsections with Tables 3-4 ($\alpha$ ablation and cross-reasoner generalization)
> - **Section 4.3**: Clarified table captions and metrics definitions
> - **Section 6**: Revised conclusion with appropriate scope, separated demonstrated results from speculative extensions
> - **Appendix A.2**: New section explaining BNPO advantage computation
> - **Appendix C**: New section with detailed dataset descriptions
> - **Appendix E**: New section with training dynamics figure
> - **Appendix F**: New section with qualitative caption comparison
>
> ---
>
> We remain available to address any further questions.
>
> Thank you,
>
> Authors

---

### Meta-Review · Area_Chair_AMp7 · 2026-01-09

**Summary:**

Key reviewer concerns include: limited adaptability due to a frozen reasoner (no co-adaptation), initially unjustified hyperparameter α, narrow math-heavy task scope with unclear transfer to perception-intensive tasks, potential overfitting to the training reasoner, counterintuitive negative clarification gap on MathVerse, WeMath performance regression with unaddressed trade-offs, and initial presentation/clarity issues (undefined jargon, missing qualitative examples/training dynamics). These unresolved core concerns outweigh the addressed points, leading to the rejection recommendation.

**Reviewer Concerns:**

Addressed: $\alpha$ ablation study (Table 3), GRPO/BNPO jargon definitions, qualitative caption examples (Appendix F), training dynamics tracking (Appendix E), preliminary cross-reasoner generalization results, Qwen-3B typo fix, Acc_deny definition clarification, and reward design misunderstanding (GRPO gradient handling).

Outstanding: Frozen reasoner constraint limiting co-adaptation (framed as practical but unaddressed as a limitation), no non-math task experiments (scope transfer remains unproven), insufficiently convincing explanation for MathVerse’s negative clarification gap, unelaborated WeMath regression trade-offs, potential residual overfitting to the training reasoner (preliminary cross-reasoner results are limited), and untested transfer to unstructured clarification tasks.

**Reviewer Scores:**

Reviewer 13NB: 4 -> 4 – $\alpha$ and adaptability framing addressed, but frozen reasoner/scope concerns persist.

Reviewer 5pEc: 2 -> 4 – $\alpha$ ablation and reward design clarified, but core concerns about supervision signal quality remain.

Reviewer cc99: 6 -> 6 – qualitative/training dynamics addressed, but MathVerse gap explanation still unconvincing.

Reviewer JaUc: 4 -> 4 – jargon/$\alpha$/cross-reasoner issues resolved, but frozen paradigm justification and scope limitations remain unaddressed.

---

### Decision · Program_Chairs · 2026-01-26

Reject